# Meta-analysis of single-cell RNA sequencing co-expression in human neural organoids reveals their high variability in recapitulating primary tissue

Jonathan M. Werner[1], Jesse Gillis[1,2]*

**1** The Stanley Institute for Cognitive Genomics, Cold Spring Harbor Laboratory, Cold Spring Harbor, New York, United States of America, **2** Physiology Department and Donnelly Centre for Cellular and Biomolecular Research, University of Toronto, Toronto, Canada

* jesse.gillis@utoronto.ca

**Data Availability Statement:** The source data and code used for all plots is made available at https://doi.org/10.5281/zenodo.13946248 (DOI:10.5281/

## Abstract

Human neural organoids offer an exciting opportunity for studying inaccessible human-specific brain development; however, it remains unclear how precisely organoids recapitulate fetal/primary tissue biology. We characterize field-wide replicability and biological fidelity through a meta-analysis of single-cell RNA-sequencing data for first and second trimester human primary brain (2.95 million cells, 51 data sets) and neural organoids (1.59 million cells, 173 data sets). We quantify the degree primary tissue cell type marker expression and co-expression are recapitulated in organoids across 10 different protocol types. By quantifying gene-level preservation of primary tissue co-expression, we show neural organoids lie on a spectrum ranging from virtually no signal to co-expression indistinguishable from primary tissue, demonstrating a high degree of variability in biological fidelity among organoid systems. Our preserved co-expression framework provides cell type-specific measures of fidelity applicable to diverse neural organoids, offering a powerful tool for uncovering unifying axes of variation across heterogeneous neural organoid experiments.

## Introduction

Pluripotent stem cells create self-organized multicellular structures, termed organoids, when cultured in a 3D in vitro environment [1,2]. The advantage of organoid models over 2D cell culture counterparts is their ability to generate structures that resemble endogenous tissues both in the differentiated cell types produced and their 3D spatial organization [3,4]. The ability to model organogenesis in a controlled in vitro environment creates opportunities to study previously inaccessible developmental tissues from both humans and a range of model organisms [5–7]. As such, organoids are genetically accessible [8] and environmentally perturbable [9] models enabling the study of molecular, cellular, and developmental mechanisms behind tissue construction. However, the applicability of studies in organoids to in vivo biology hinges

zenodo.13946248) as well as at https://github.com/JonathanMWerner/meta_organoid_analysis. The GitHub repository contains an easy to view GitHub markdown file containing code for generating all figure plots at https://github.com/JonathanMWerner/meta_organoid_analysis/blob/main/figure_plots_with_data_code.md. Due to file size limits on GitHub, several source data files are only available in the Zenodo repository, detailed in the Zenodo repository description. The preservedCoexp R library is made available at https://github.com/JonathanMWerner/preservedCoexp.

**Funding:** JG and JMW were supported by National Institutes of Health grants R01MH113005 and R01LM012736. JMW was supported by National Science Foundation award no. DGE-1938105. The funders did not play any role in the study design, data collection and analysis, decision to publish, or preparation of the manuscript. National Institutes of Health URL: https://www.nih.gov/ National Science Foundation URL: https://www.nsfgrfp.org/.

**Competing interests:** The authors have declared that no competing interests exist.

**Abbreviations:** AUROC, area-under-the-receiver-operating-characteristic curve; DE, differentially expressed; ECM, extra-cellular matrix; IQR, interquartile range; PCA, principal component analysis.

on how well these in vitro models recapitulate primary tissue developmental processes, which remains an open question.

Quantifying the degree to which organoid systems replicate primary tissue biological processes is a critical step toward understanding the strengths and limitations of these in vitro models [10–14]. However, studies that perform such primary tissue/organoid comparisons are inherently confounded by batch [15] (in vivo versus in vitro), making it difficult to disentangle batch effects from underlying primary tissue and organoid biology. Meta-analytic approaches across many primary tissue and organoid data sets offer a route around these confounds, enabling the discovery of replicable primary tissue and organoid signatures independent of batch, which can then be interrogated for how well organoids recapitulate primary tissue biology [16–18]. A useful biological signature for this purpose is gene co-expression [19]. Genes that are functionally related tend to be expressed together, resulting in correlated gene expression dynamics that can define functionally relevant gene modules [19]. Gene co-expression relationships represent a shared genomic space that can be aggregated across experiments (e.g., [20]) in either in vivo or in vitro systems, thus providing a useful framework for quantifying functional similarities and differences. Excitingly, coupling meta-analytic comparisons of primary tissue and organoid co-expression with single-cell RNA-sequencing data (scRNA-seq) stands to deliver cell type-specific quantifications of organoids' current capacity for producing functionally equivalent cell types to primary tissues [21,22].

Among organoid systems, human neural organoids are particularly well suited for meta-analytic evaluation due to well-described broad cell type annotations and their known lineage relationships [23], the wide variety of differentiation protocols in use [24], and the increasing amount of single-cell primary brain tissue and neural organoid data publicly available. In particular, the diversity of differentiation protocols for human neural organoids poses a unique challenge for organoid quality control that can be met by meta-analytic approaches. Neural organoids can either be undirected [25] (multiple brain region identities) or directed (specific brain region identity) with an increasing number of protocols striving to produce a wider variety of region-specific organoids [11,26–37]. Meta-analytic primary tissue/organoid comparisons across differentiation protocols stand to derive generalizable quality control metrics applicable to any differentiation protocol, fulfilling a currently unmet need for unified quality control metrics across heterogeneous neural organoids.

Prior comparisons between primary brain tissues and neural organoids demonstrated that organoids have the capacity to produce diverse cell types that capture both regional and temporal variation similar to primary tissue data as assayed through transcriptomic [10,11,13,16,17,38], epigenomic [39,40], electrophysiologic [41], and proteomic studies [42]. At the morphological level, neural organoids can produce cellular organizations structurally similar to various in vivo brain regions, including cortical layers [43] and hippocampus [27], as well as modeling known inter-regional interactions like neuromuscular junctions [34] and interneuron migration [29]. Additionally, several prior studies have compared primary tissue/organoid co-expression and concluded that neural organoids recapitulate primary brain tissue co-expression [5,13,39], but these assessments are highly targeted to study-specific properties, limiting potential generalization or potential assessment across the field. Typically, only a single organoid differentiation protocol is used in these assessments and it remains unclear whether organoids across different protocols will produce similar results. This lack of breadth also affects the use of primary tissue data used as a reference, with the primary tissue data sets utilized being treated as gold standard data sets with little consideration for the extent one primary tissue reference may generalize to another. Furthermore, while prior meta-analytic comparisons of primary tissue and organoids have been conducted that include different neural organoid protocols, these analyses were either performed at the bulk level [17] without cell

type resolution or relied on single-cell integration approaches [44,45], thereby limiting the biological resolution or scalability of the findings.

In this study, we perform a meta-analytic assessment of primary brain tissue (2.95 million cells, 51 data sets; Fig 1A and S1 Table) [46–54] and neural organoid (1.59 million cells, 173 data sets, 10 neural organoid, and 2 non-neural organoid protocols; Fig 1B and S1 Table) [6,8,11,12,29,31,33,35,36,38,41,43,44,55–69] scRNA-seq data sets, constructing robust primary tissue cell type-specific markers and co-expression to query how well neural organoids recapitulate primary tissue cell type-specific biology. We sample primary brain tissue data over the first and second trimesters and across 15 different developmentally defined brain regions, extracting lists of cell type markers that define broad primary tissue cell type identity regardless of temporal, regional, or technical variation (Fig 1A). We derive co-expression networks from individual primary tissue and organoid data sets as well as aggregate co-expression networks across data sets (Fig 1C). From these networks, we assess the strength of co-expression within primary tissue cell type marker sets as well as the preservation of co-expression patterns between primary tissue and organoid data (Fig 1D and 1E). We also provide an R package to download our primary tissue reference co-expression network to assay new neural organoid data using simple, meaningful, and fast statistics (Fig 1F). By constructing robust primary tissue cell type representations through meta-analytic approaches, we demonstrate the preservation of primary tissue cell type co-expression provides both specific and generalizable characterization of the primary tissue fidelity of human neural organoids.

## Results

### Meta-analytic framework for primary tissue/organoid comparisons

We reason that, if they exist, primary tissue cell type-specific signals robust to temporal, regional, and technical variation will constitute in vivo standards applicable to any organoid data set regardless of time in culture or differentiation protocol. We first show it is possible to learn sets of marker genes that define broad primary tissue cell types (Fig 1A and S2 Table) across time points (gestational weeks GW5-GW25) and brain regions (15 developmentally defined brain regions) through a meta-analytic differential expression framework (Figs 1A, 2A and 2B). We then compare co-expression within these marker sets between primary tissue and organoid data to quantify the degree organoids preserve primary tissue cell type-specific co-expression. An important aspect of our analysis is our cross-validation of primary tissue differential expression and co-expression. We employ a leave-one-out cross-validation approach when learning robust differentially expressed marker genes from our annotated primary tissue data sets (2,174,934 cells, 37 data sets) and we interrogate co-expression of our primary tissue marker genes within a large cohort of unannotated primary tissue data sets (776,344 cells, 14 data sets). This approach ensures we are extracting primary tissue markers and co-expression relationships independent of temporal, regional, and technical variation, a powerful approach for deriving broad primary tissue signatures appropriate for comparison to a wide range of organoid data sets.

### Cross-temporal and -regional primary tissue cell type markers

To learn markers that define broad primary tissue cell types (see Methods), we apply the Meta-Markers [70] framework to our cross-temporal and -regional annotated primary tissue data sets (Fig 2A and 2B). MetaMarkers uses robust differential expression statistic thresholds (log2 fold-change $> = 4$ and FDR-adjusted $p$-value $< = 0.05$) for determining whether a gene is differentially expressed (DE) within individual data sets, then ranks all genes via the strength of their recurrent DE across data sets (see Methods). We test the generalizability of our primary

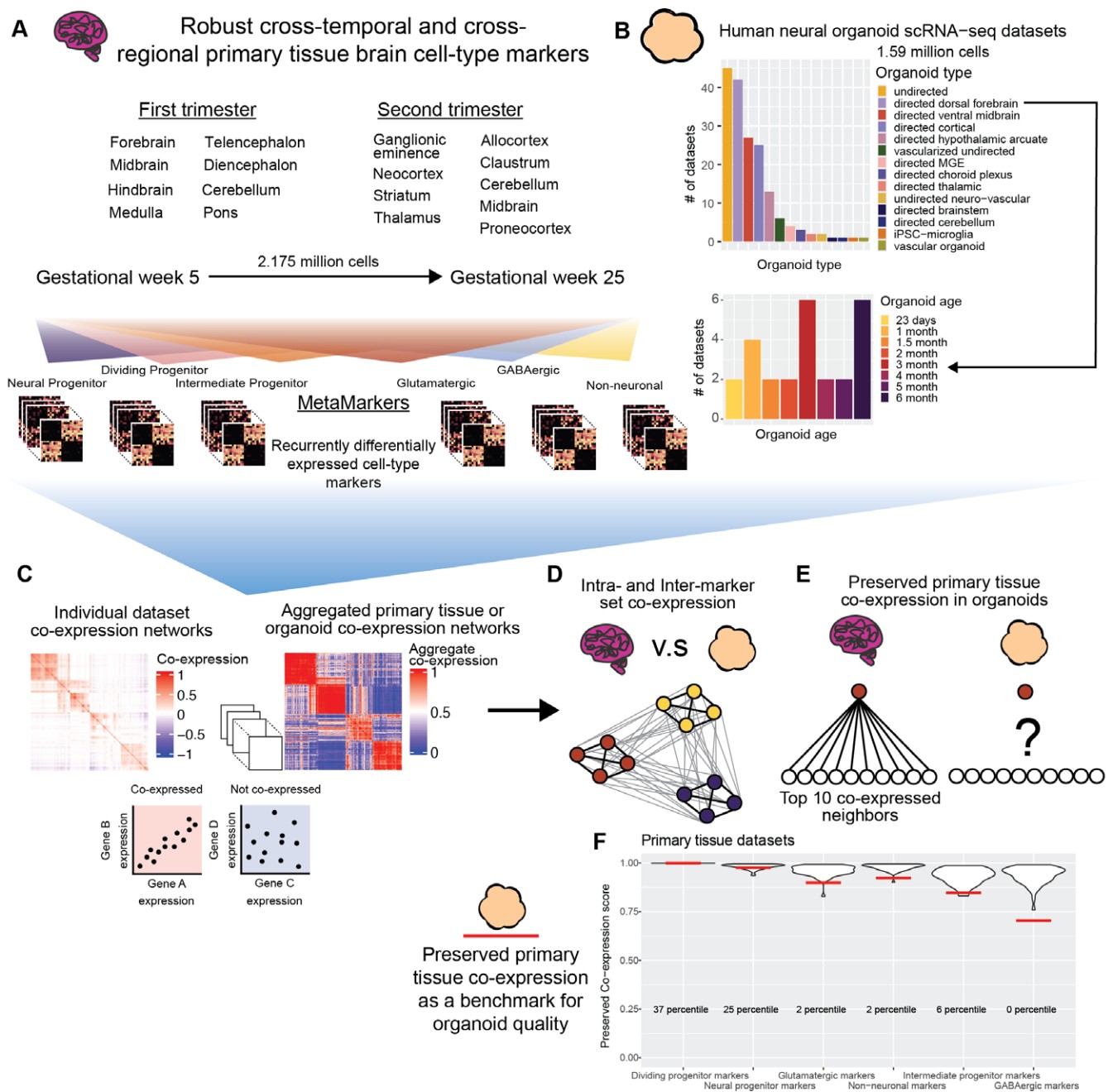

**Fig 1. Using meta-analysis to quantify preserved primary tissue co-expression in organoids.** (A) Collection of annotated primary tissue brain scRNA-seq data sets, ranging from gestational week (GW) 5 to 25 and sampling from 15 developmentally defined brain regions. The primary tissue data sets are annotated at broad cell type levels (Neural Progenitor, Dividing Progenitor, Intermediate Progenitor, Glutamatergic, GABAergic, and Non-neuronal) and these annotations are used to compute MetaMarkers, cell type markers identified through recurrent differential expression. (B) Collection of human neural organoid scRNA-seq data sets, sampling from 12 different differentiation protocols. We also include vascular organoid and iPSC-microglia data sets as non-neural organoid controls. Included is a cell type annotated temporal forebrain organoid data set. (C) Example of a sparse co-expression network derived from scRNA-seq data and an example of an aggregate co-expression network averaged over many scRNA-seq data sets. The aggregate network enhances the sparse signal from the individual network. (D) Schematic showing a quantification of intra- and inter-marker set co-expression. (E) Schematic showing a quantification for the strength of preserved co-expression between 2 co-expression networks, measuring the replication of the top 10 co-expressed partners of an individual gene across the networks. (F) Example plot from the preservedCoexp R library, placing cell type-specific preserved co-expression scores of an example forebrain organoid data set in reference to scores derived from primary tissue data sets. Red lines denote the percentile of the organoid cell type scores within the primary tissue distributions. Underlying data can be found in the Zenodo repository (doi:10.5281/zenodo.13946248).

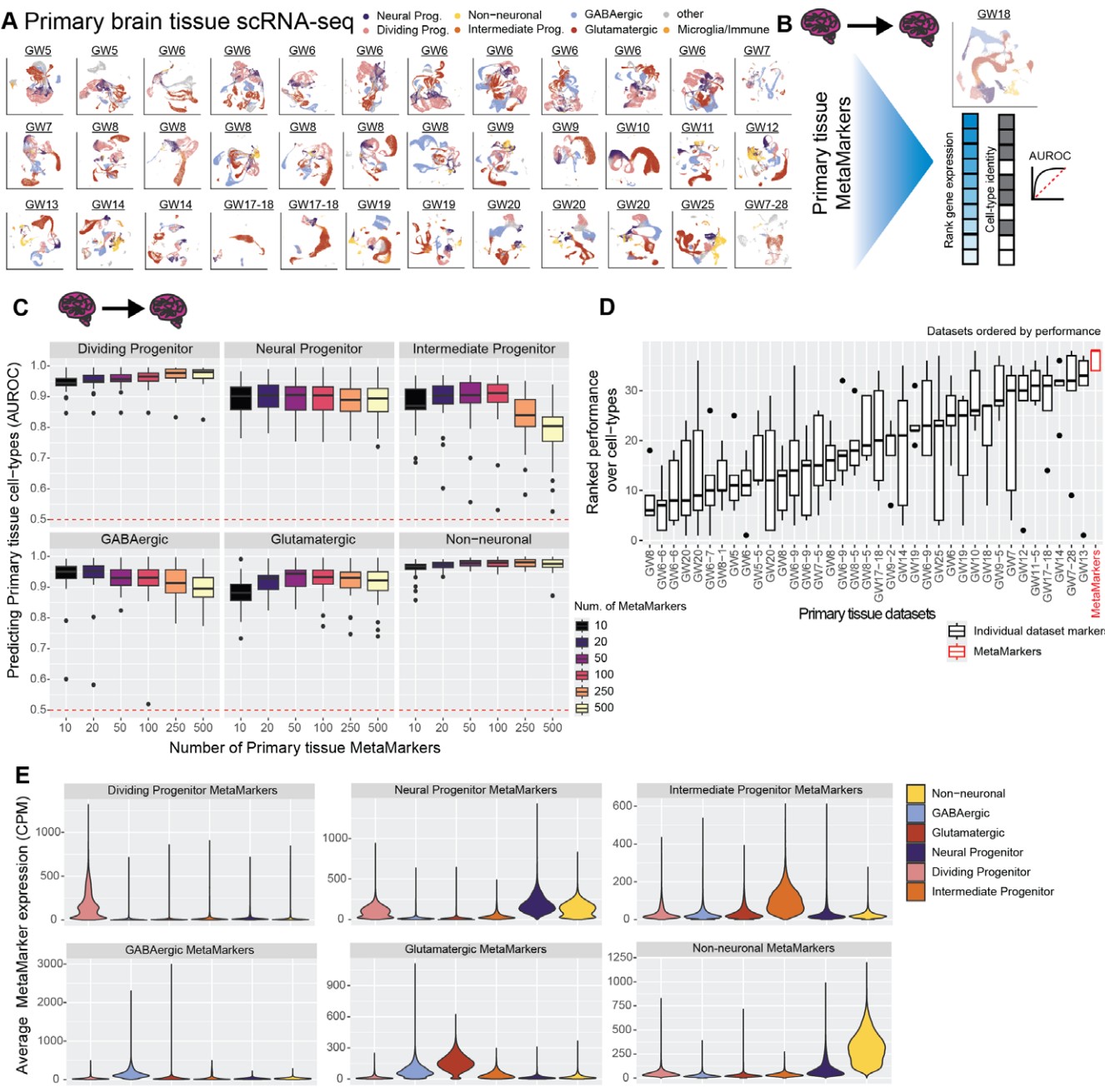

**Fig 2. Meta-analytic primary tissue cell type markers. (A)** Annotated UMAPs of the annotated primary tissue brain scRNA-seq data sets. **(B)** Example of our leave-one-out cross-validation approach for learning primary tissue MetaMarkers and testing the markers' capacity for predicting annotations in the left-out data set, quantified with the AUROC statistic. **(C)** Meta-analytic primary tissue markers have high performance in predicting primary tissue cell type annotations. Boxplot distributions of the AUROC statistic for predicting cell type annotations across all leave-one-out combinations of our annotated primary tissue data sets, with an increasing number of MetaMarkers used for predicting cell type annotations on the x-axis. **(D)** MetaMarkers have the highest performance in predicting primary tissue cell type annotations compared to data set-specific markers. Boxplots of marker gene set performances. Gene sets are the top 100 cell type markers from individual primary tissue data sets compared to the top 100 MetaMarkers performance. Performances for each cell type in individual primary tissue data sets are presented in S1A Fig. Data sets are ordered by their median performance. **(E)** Averaged distributions of gene expression for the top 100 MetaMarkers. This is performed with a leave-one-out cross-validation, with individual data set distributions reported in S1B Fig. Underlying data can be found in the Zenodo repository (doi:10.5281/zenodo.13946248).

tissue MetaMarker gene sets in predicting primary cell types by employing a leave-one-out primary tissue cross-validation (Fig 2A and 2B). We construct an aggregate expression predictor in the left-out data set using MetaMarkers learned from the remaining data sets (see Methods), quantifying how well the MetaMarker gene sets predict the left-out cell type annotations with the area-under-the-receiver-operating-characteristic curve statistic (AUROC; Fig 2B and 2C). The AUROC is the probability of correctly prioritizing a true positive (e.g., cell of the right type) above a negative, (e.g., cell of the wrong type), given some predictor of the positive class, in this case, aggregate cell type marker expression.

Starting with just the top 10 primary tissue MetaMarkers per cell type, we achieve a median AUROC across all primary tissue data sets of 0.945 (interquartile range (IQR): 0.935–0.959), 0.901 (IQR: 0.858–0.931), 0.870 (IQR: 0.858–0.925), 0.950 (IQR: 0.928–0.967), 0.882 (IQR: 0.857–0.909), and 0.967 (IQR: 0.958–0.975), for dividing progenitors, neural progenitors, intermediate progenitors, GABAergic neurons, glutamatergic neurons, and non-neuronal cell types (oligodendrocytes, oligodendrocyte precursor cells, and astrocytes), respectively (Fig 2C). These extremely high performances demonstrate that even a small number of meta-analytically derived primary tissue cell type markers have high utility in predicting primary tissue cell type annotations regardless of temporal and regional variability. We provide a list of the top-ranked MetaMarkers per cell type in S3 Table. For all following analysis, we take the top 100 MetaMarkers per cell type as robust representations of our 6 broad primary tissue cell type annotations (median AUROCs range from 0.904 to 0.980), with the 100 MetaMarkers achieving modest increases in performance over the top 10 MetaMarkers for all cell types except GABAergic cells (Fig 2C, median AUROC for 100 GABAergic MetaMarkers: 0.931 IQR: 0.905–0.957). When comparing MetaMarkers to markers derived from individual primary tissue data sets, we find the MetaMarkers are consistently top performers in predicting primary tissue annotations (Fig 2D), with MetaMarkers producing the top results for intermediate progenitors, glutamatergic neurons, GABAergic neurons, and non-neurons (S1 Fig), as well as comparable performance to top individual data sets for dividing and neural progenitors (S1 Fig).

We explore the primary tissue MetaMarker sets further by computing the average expression of the top 100 MetaMarkers for our 6 annotated cell types across all cells within our 37 annotated primary tissue data sets (Fig 2E), continuing our leave-one-out approach. Each annotated primary tissue cell type expresses the corresponding matched MetaMarker set over all other MetaMarker sets, with the exception of off-target expression for the neural progenitor MetaMarkers in astrocytes (aggregated over all data sets Fig 2E, individual data sets S1B Fig). This demonstrates our MetaMarker gene sets act as robust cell type markers in aggregate across all first and second trimester time points (Figs 2E and S1B). Additionally, we investigate the expression of the top 100 MetaMarker gene sets across annotated primary brain regions (S2A and S2B Fig). While each primary tissue cell type predominantly exhibits maximal expression of the corresponding primary tissue MetaMarker set across brain regions compared to other marker sets, there is shared expression of the neural progenitor MetaMarkers for the non-neurons and neural progenitors that appears in a data set-specific manner (S2A and S2B Fig). Overall, we are able to meta-analytically extract cell type markers that define broad primary tissue cell types independent of temporal and regional variation.

## Broad primary tissue cell type markers capture organoid temporal variation

After extracting meta-analytic cell type markers that capture broad primary tissue temporal and regional variation, we can test how well these markers also capture organoid temporal and

regional (protocol) variation. We start with a large-scale temporal organoid atlas with cell type annotations [38] derived from a forebrain differentiation protocol containing time points ranging from 23 days to 6 months in culture. When comparing primary tissue and organoid data along a temporal axis, one might expect younger primary tissue expression data to be a better reference for younger organoid cell types (better able to predict cell-types) and vice-versa for older primary and organoid data (S3A Fig). We test this relationship using the same AUROC quantification as in Fig 1C, but now using the top 100 primary tissue cell type markers per primary tissue data set to predict organoid cell type annotations across all organoid time points (S3B Fig, see Methods).

We observe highly consistent performance across all primary tissue data sets (GW5–GW25) when predicting organoid cell types regardless of the organoid time point (S3B Fig). The average absolute difference in AUROC scores when predicting organoid cell types using either our youngest (GW5) or oldest (GW25) primary data is 0.000382 ± 0.0357 SD, 0.141 ± 0.192 SD, 0.0139 ± 0.0317 SD, 0.00171 ± 0.113 SD, and 0.0712 ± 0.0607 SD for dividing progenitors, neural progenitors, glutamatergic neurons, GABAergic neurons, and non-neuronal cells, respectively (no annotated intermediate progenitors for several older primary tissue data sets). This demonstrates strikingly consistent performance across distant primary tissue time points, highlighting that broad primary tissue cell type signatures are applicable as reference for organoid cell types regardless of the primary tissue or organoid time point. The one possible exception is for neural progenitors, where there seemingly is a temporal shift in performance with younger primary tissue data sets predicting younger organoid annotations over older organoid annotations and vice-versa for older primary tissue/organoid data (S3B Fig). However, a subset of the young GW6-8 primary tissue data sets report sharp increases in performance predicting older organoid time points in opposition to other GW6-8 primary tissue data sets, suggesting variance in performance is driven by intersections of technical variability between individual organoid and primary tissue data sets rather than overarching temporal variability. Importantly, our lists of top 100 primary tissue MetaMarkers produce the highest mean AUROC across the 6 cell types in comparison to markers from individual data sets (MetaMarker mean AUROC across cell types: 0.887, individual data set means range from 0.751 to 0.867, S3B Fig). This demonstrates our meta-analytic primary tissue cell type markers capture organoid temporal variation more generally across cell types than any individual primary tissue data set.

## Broad primary tissue cell type markers characterize organoid expression variability

Due to the lack of publicly available cell type annotations for the vast majority of the organoid data sets we sample, we investigate whether our primary tissue MetaMarker gene sets characterize organoid variability through principal component analysis (PCA). Genes that are heavily weighted in the first PC of a transcriptomic data set are sources of significant variability in gene expression and likely represent important biological variability; this is largely the reasoning for using PCA as a feature selection tool in standard transcriptomic analyses. We find that our lists of 100 primary tissue MetaMarkers have higher PC1 weights (absolute-value of PC1 weights that are then min-max normalized per data set) than non-marker genes and are consistently heavily weighted in PC1 across organoid data sets more so than non-marker genes (S3C and S3D Fig). This demonstrates our MetaMarker gene sets characterize greater sources of gene expression variability across organoid data sets compared to non-marker genes.

## Quantifying intra- and inter-MetaMarker co-expression of neural organoids

Our primary tissue MetaMarkers that represent both primary tissue and organoid temporal/regional variation enable assessments of cell type-specific co-expression between arbitrary primary tissue and organoid data sets. One normally would need matched cell type annotations across data sets to compare cell type-specific biology, but here we couple our meta-analytically derived cell type markers with gene co-expression quantifications, which do not rely on cell type annotations, to extract cell type-specific co-expression from any given scRNA-seq data set. Practically, if organoids are producing cell types functionally identical to primary tissue cell types, we would expect near identical co-expression relationships of the primary tissue MetaMarker gene sets across primary tissue and organoid data sets.

We first visualize marker set co-expression within our unannotated primary tissue (not included in generating the primary tissue MetaMarker gene sets) and neural organoid data sets through aggregate co-expression networks, which provide well-powered summarizations of co-expression relationships present within the data (Fig 3A and 3B, see Methods). While intra-marker set co-expression qualitatively appears comparable between the aggregate networks, the organoid network contains noticeably increased inter-marker set co-expression, particularly for the glutamatergic and GABAergic markers. To explore further, we quantify both the intra- and inter-marker set co-expression of MetaMarker gene sets for individual and aggregate networks.

We score intra-gene set co-expression strength through a simple machine learning framework [71,72], which quantifies whether genes in a given set are more strongly co-expressed with each other compared to the rest of the genome (Fig 3C). Co-expression module scores across the annotated and unannotated primary tissue data sets are largely comparable with the exception of a sharp decrease in intermediate progenitor performance for the unannotated primary tissue data sets (Fig 3D). Six out of the 14 unannotated data sets are sampled from either the ganglionic eminences or the hypothalamus, potentially explaining this decrease in performance and suggesting our intermediate progenitor MetaMarkers are enriched for signal from cortical areas. In contrast, performance is much more variable across the individual organoid data sets for all cell types except the dividing progenitors, ranging from no signal (AUROC ~ 0.50) to comparable results with primary tissue networks (Fig 3D). Notably, the intra-marker set co-expression of the aggregate organoid network is nearly identical to the aggregate primary tissue networks (diamond, triangle, and square special characters, Fig 3D). This demonstrates that while individual organoid data sets may suffer in performance compared to primary tissue, neural organoids in aggregate broadly replicate co-expression relationships within primary tissue marker sets.

The variation among individual organoid data sets for co-expression of the primary tissue cell type markers may be influenced by the compositional variation in cell types produced across the differentiation protocols. A protocol that aims to produce a directed excitatory lineage organoid is not expected to produce inhibitory cell types and thus should not necessarily exhibit strong inhibitory neuron co-expression. We test this relationship by first estimating the cell type compositions of each neural organoid data set through their expression of the primary tissue MetaMarkers. Briefly, cells are scored for their aggregate expression of genes within each MetaMarker gene set (using the top 15 MetaMarkers per cell type) and then assigned a cell type label determined by the MetaMarker gene set with the highest enriched expression. Comparing this annotation approach to author-provided annotations for the temporal directed forebrain neural organoid atlas reveals high concordance in cell type annotations (S4A Fig), with some misannotations between neural progenitors and non-neurons (S4B

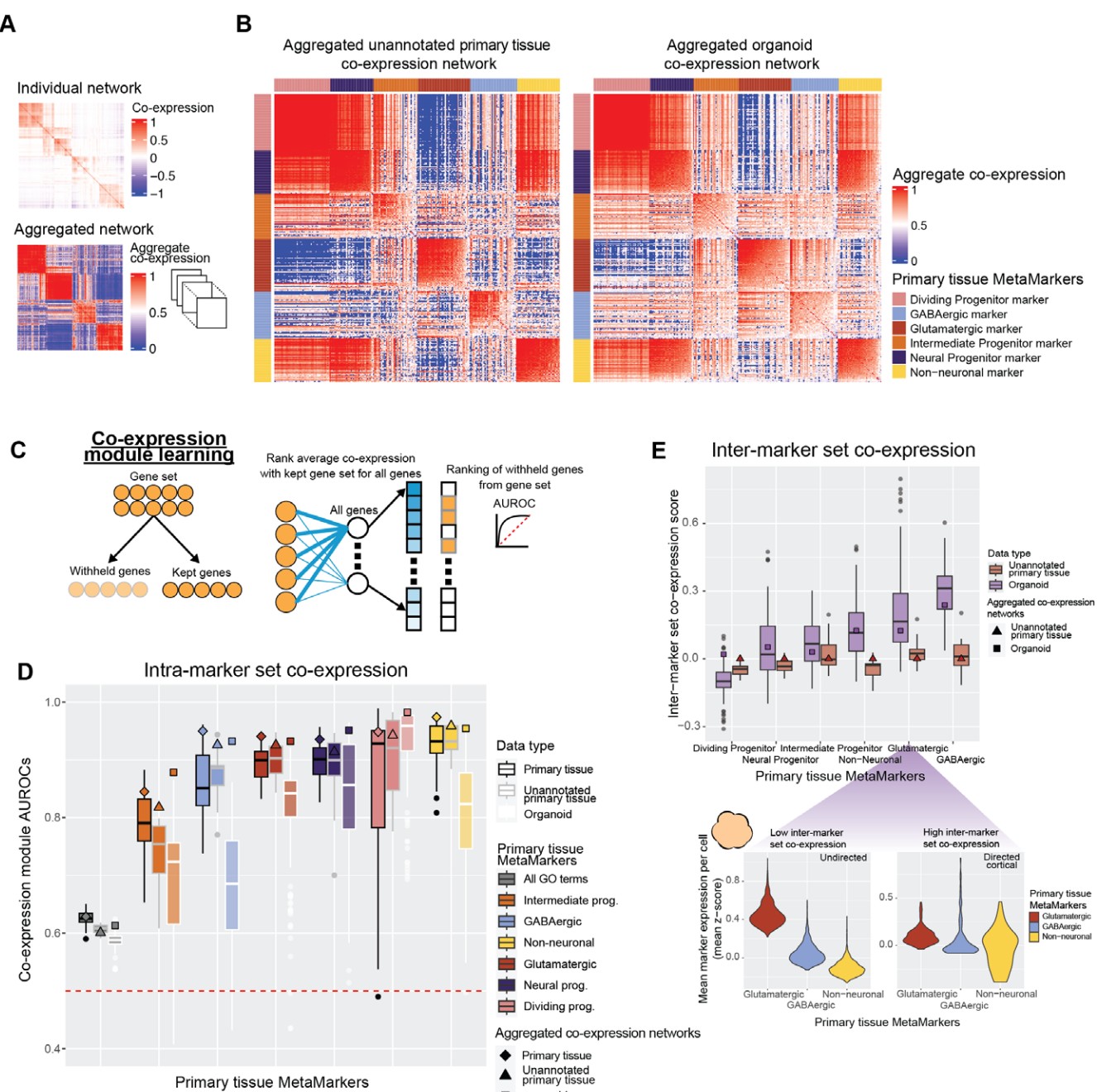

**Fig 3. Neural organoids vary in recapitulating primary tissue cell type marker set co-expression. (A)** Example of a sparse co-expression network derived from scRNA-seq data and an example of an aggregate co-expression network averaged over many scRNA-seq data sets. The aggregate network enhances the sparse signal from the individual network. **(B)** The aggregated co-expression networks for the unannotated primary tissue data sets and organoid data sets. Genes are grouped by MetaMarker gene sets and ordered within gene sets by their average intra-gene set co-expression. **(C)** Schematic for the co-expression module learning framework, measuring the co-expression strength within an arbitrary gene set compared to the rest of the genome, quantified with the AUROC statistic. **(D)** Distributions of co-expression module AUROCs for individual annotated primary tissue, unannotated primary tissue, and organoid data sets for the co-expression strength of the MetaMarker gene sets for the 6 cell types. The gray "All GO terms" distributions report the average co-expression module AUROC across all GO terms for each individual data set. Co-expression module AUROCs for the aggregate co-expression networks are denoted with the special characters. **(E)** Distributions of inter-marker set co-expression ratios standardized to the ratios of the aggregate unannotated primary tissue network (triangle special character). The bottom plots show violin plot distributions of MetaMarker expression of the bottom (left, low inter-marker set co-expression) and top (right, high inter-marker set co-expression) scoring organoid data sets for glutamatergic inter-marker set co-expression. The violin plots depict the mean z-score of the corresponding MetaMarkers within each cell, reported for the top 10% of cells per data set determined via glutamatergic MetaMarker expression. The protocol type is labeled in the top right corner for each panel. Underlying data can be found in the Zenodo repository (doi:10.5281/zenodo.13946248).

Fig), again most likely attributable to the shared expression of the neural progenitor MetaMarkers with astrocytes. We then group neural organoid data sets by their shared cell type percentages and compare their co-expression module scores, revealing pervasively weak relationships between a data set's cell type percentage and its associated co-expression module score, particularly for the differentiated cell types (S4C and S4D Fig).

Starting with glutamatergic composition, the range of co-expression module scores for data sets with 0% to 10% glutamatergic cells is 0.399 to 0.920, which spans from no signal (~0.5) to signal comparable to primary tissue scores (~>0.85, S4D Fig). Organoid data sets with greater than 10% glutamatergic cells (spanning >10% to 90%) have scores ranging from 0.560 to 0.911, exceedingly similar to data sets with 10% or less glutamatergic cells. This demonstrates organoids with small glutamatergic percentages (<10%) are producing glutamatergic co-expression relationships equal to data sets with much higher glutamatergic percentages (10% to 90%). The GABAergic, non-neuronal, and intermediate progenitor cell type compositions display the same trend when comparing the lowest cell type percentage data sets to all other data sets (GABAergic: [0%–10%] scores 0.312–0.826, (10%–90%] scores 0.440–0.862. Non-neuronal: [0%–10%] scores 0.497–0.929, (10%–60%] scores 0.683–0.948. Intermediate Progenitors: [0%–5%] scores 0.379–0.826, (5%–20%] scores 0.371–0.822, S4D Fig). The neural and dividing progenitors are the exception, with higher estimated cell type percentages trending with lower co-expression module scores, which is more apparent with the neural progenitors than the dividing progenitors (S4D Fig). This is most likely due to the misannotations between neural progenitors and non-neurons (likely astrocytes). In general, the majority of organoid data sets with near-zero co-expression signal are also the data sets with near-zero predicted composition of the respective cell type, particularly for the differentiated cell types (S4D Fig). This suggests co-expression performance within neural organoids is influenced largely on whether or not the cell type is present within the organoid, regardless of the exact prevalence of the cell type. As another example of this broad compositional effect on co-expression, we compare the co-expression module scores of all neural lineage MetaMarkers to scores for Microglia/Immune MetaMarkers (derived identically as the other primary tissue MetaMarkers, see Methods, S3 Table) between the neural organoid protocols and the 2 non-neural organoid protocols we sample (vascular organoid and iPSC-microglia data sets), which are not expected to produce neural lineage cell types. The non-neural organoid data sets have near-zero signal for intra-marker set co-expression of the neural lineage MetaMarkers (co-expression module score ~0.50), while having high scores for the microglia/immune MetaMarkers (co-expression module score ~0.90), with the neural organoids exhibiting the exact opposite results (S5A Fig). Taken together, our results show neural organoids ranging from very small to very large percentages of a given cell type produce comparable intra-marker set co-expression signal, while organoids that do not produce a given cell type have near-zero co-expression signal of the corresponding cell type markers, most likely simply explained by the lack of expression of those cell type markers. To further explore co-expression variability of expressed genes between primary tissue and neural organoids, we turn to scoring inter-marker set co-expression.

We quantify inter-marker set co-expression by scoring the ratio of average inter-marker set to intra-marker set co-expression of individual MetaMarker genes for all co-expression networks, taking the average ratio within each MetaMarker gene set to summarize individual networks. We then standardize to the ratios of the aggregate unannotated primary tissue network (Fig 3E, raw ratios in S5B Fig). Broadly, neural organoids produce higher inter-marker set co-expression compared to primary tissue data sets, particularly for the differentiated cell types (non-neuronal, glutamatergic, and GABAergic MetaMarker sets). As an example, we visualize MetaMarker expression of the bottom and top scoring neural organoid data sets for

glutamatergic inter-marker set co-expression, considering only the top 10% of cells that express glutamatergic MetaMarkers within each data set. This demonstrates the range of inter-marker set co-expression across neural organoids, which vary from cells with nearly exclusive expression of glutamatergic MetaMarkers to cells that express markers from glutamatergic, GABAergic, and non-neuronal MetaMarker gene sets (Fig 3E). Taking the intra- and inter-marker set co-expression quantifications together, our results demonstrate that while neural organoids can produce comparable intra-marker set co-expression to primary tissue; this is accompanied by extensive off-target cell type marker co-expression.

## Organoid data sets vary in preserving gene-level primary tissue co-expression

We take our primary tissue/organoid co-expression comparisons a step further and ask how well individual organoid data sets preserve gene-level primary tissue co-expression relationships. For any given individual gene, we quantify whether that gene's top co-expressed partners are preserved in one co-expression network compared to another (Fig 4A). We use the aggregate co-expression network from the annotated primary tissue data sets as our reference co-expression network and test how well individual co-expression networks, either primary tissue or organoid, perform in preserving primary tissue gene-level co-expression patterns (Fig 4A, top 10 co-expressed neighbors). We start by quantifying the preserved co-expression of genes within our primary tissue MetaMarker gene sets, using the average preserved co-expression AUROC as a measure of preserved co-expression for any given gene set (Fig 4A). Across our 6 annotated primary tissue cell types, primary tissue co-expression networks deliver consistently high performance for preserved co-expression scores of our primary tissue MetaMarker gene sets (Fig 4B, mean preserved co-expression score across cell types and primary tissue data sets: annotated 0.970 ± 0.0229 SD, unannotated 0.962 ± 0.0102 SD). This indicates that across the highly temporally and regionally diverse primary tissue data, the co-expression relationships of our MetaMarker gene sets are incredibly highly preserved, again reflecting the temporally and regionally robust nature of our primary tissue cell type markers.

In contrast, organoid data sets vary substantially in preserved co-expression scores across our primary tissue MetaMarker gene sets, with exceedingly small overlaps in scores to primary tissue data sets for all MetaMarker gene sets except dividing progenitors (Fig 4B). As before with our quantification of intra-gene set co-expression, compositional variation across organoid protocols may influence the variation in performance across cell types/data sets. We again compare the preserved co-expression scores across data sets grouped by cell type composition and find the same trend as before with the intra-marker set co-expression module scores; organoids ranging from small to large cell type percentages produce largely similar preserved co-expression scores (S5C Fig). We quantify variability in preserved co-expression across neural organoids in a slightly different manner by computing the correlation in preserved co-expression between all MetaMarker gene sets, revealing significant positive correlations across all cell type marker sets in neural organoids (Spearman correlations range from 0.426 to 0.886, FDR-adjusted $p$-values are all <0.001, Fig 4C). This indicates preserved primary tissue co-expression is a global feature of organoid data sets, which is not expected if variability in cell type composition among neural organoids was a strong determinant of preserved co-expression. Importantly, regardless of the cell type composition of neural organoids, we demonstrate the vast majority of the sampled organoids consistently fail to preserve primary tissue co-expression at a level comparable to primary tissue for all the broad cell types except dividing progenitors (Fig 4B, exceedingly small overlap in the primary tissue and organoid

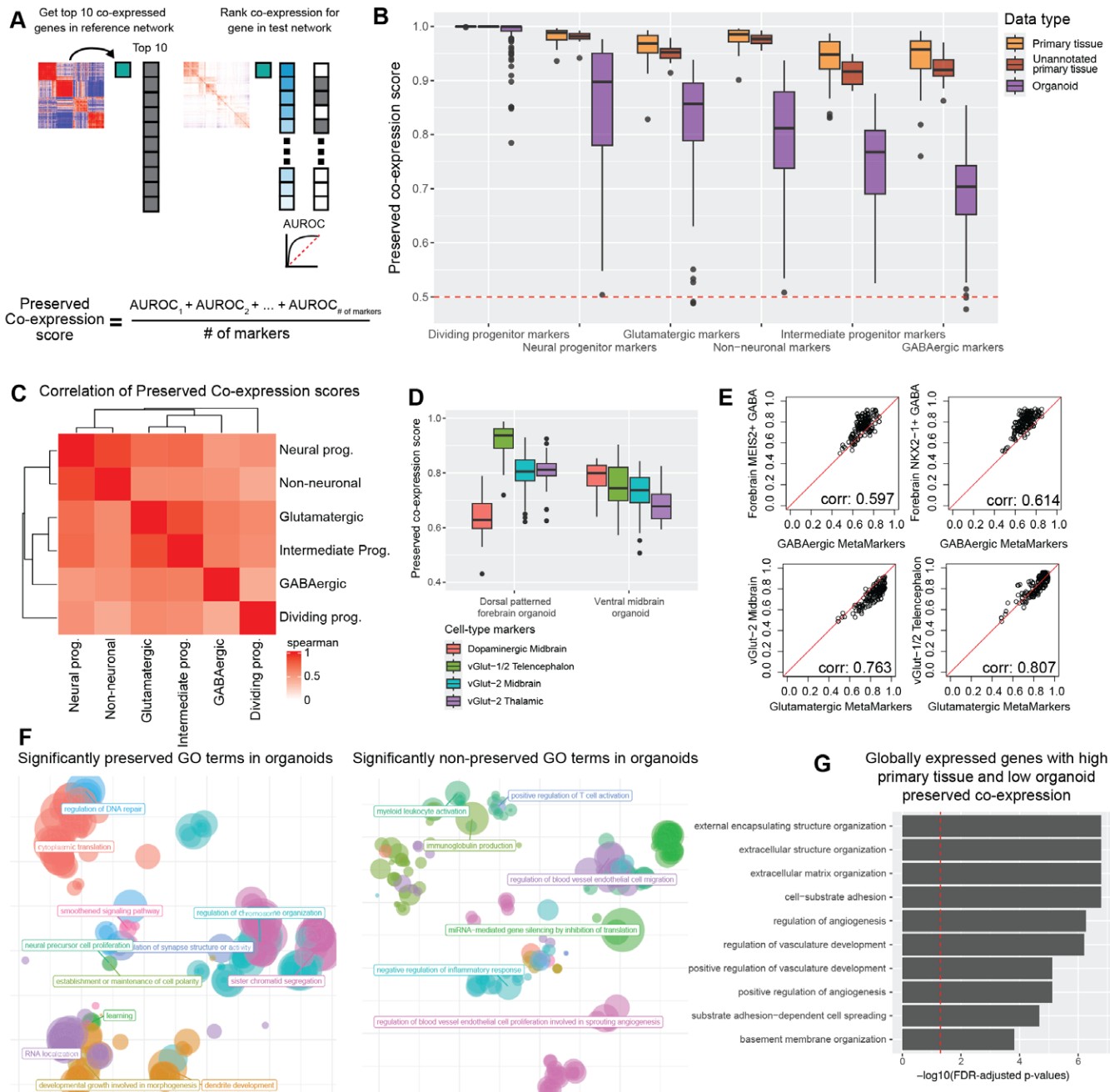

**Fig 4. Neural organoids vary in their preservation of primary tissue gene-level co-expression.** (A) Schematic showing the quantification for gene-level preserved co-expression. The preserved co-expression score for any given gene set is the average preserved co-expression AUROC across all genes within that gene set. (B) Boxplot distributions of the preserved co-expression scores for the 6 primary tissue MetaMarker gene-sets across all individual primary tissue and organoid networks. (C) Spearman correlation matrix for the preserved co-expression scores for all 6 cell type annotations across all individual organoid data sets. (D) Preserved co-expression scores computed from the directed dorsal forebrain and directed ventral midbrain organoid data sets for the top 10 cell type markers of various region-specific neural cell types. (E) Scatter plots comparing the preserved co-expression score of the top 100 MetaMarkers against the top 10 markers (no overlaps in gene sets) for various neural cell types per organoid data set. Spearman correlation coefficients are reported in the bottom right corner. (F) Scatter plots summarizing the semantic distances of GO terms that are significantly preserved or non-preserved between the aggregate annotated primary tissue and organoid co-expression networks. (G) Organoids globally fail to preserve primary tissue co-expression of ECM and vascular related genes. Bar plot detailing the top 10 GO terms from a GO enrichment test of the 98 genes with high and low preserved co-expression AUROCs within primary tissue networks and organoid networks, respectively. The preserved co-expression AUROC for each individual gene from primary tissue networks and organoid networks is reported in S8B and S8C Fig. Underlying data can be found in the Zenodo repository (DOI:10.5281/zenodo.13946248).

distributions). This suggests a persistent remaining deficit in the fidelity of neural organoids in reference to primary tissue that is independent of variability in cell type composition.

We test the robustness of our results demonstrating consistent deficits in the preservation of primary tissue co-expression among organoid data sets by repeating the cross-validation of primary tissue preserved co-expression using brain region-specific co-expression networks (S6 Fig). While we report sizable variability in preserved co-expression across brain regions for intermediate progenitors, glutamatergic, and GABAergic neurons, these scores track well with the number of cells captured per brain region, suggesting data set size is a significant contributor to performance variability across data sets (S6 Fig). Importantly, when comparing these region-specific primary tissue scores to neural organoid scores and controlling for data set size and gene detection rates, we again observe consistent deficits in performance across neural organoid data sets for all 6 cell types (S7 Fig), in agreement with our earlier results.

Neural organoids are commonly employed for the study of diverse disease mechanisms through various perturbations. We tested the relevance of our co-expression scores for quantifying primary tissue fidelity across normal and perturbed organoids. A subset of our organoid data sets come from studies that performed diverse perturbations (22q11.2 deletion, SMARCB1 knockdown, exposure to Alzheimer's serum, amyotrophic lateral sclerosis patient-derived organoids). We compare the MetaMarker co-expression scores between normal and perturbed organoids and find only 2 significant differences across all cell type MetaMarker sets (co-expression module scores normal versus mutant FDR-adjusted $p$-values: glutamatergic- 0.00544, dividing progenitor 0.0481, S8A Fig). This demonstrates our broad primary tissue cell type co-expression signatures are also applicable for comparison with organoids in perturbation experiments.

## Fidelity of region-specific cell types through preserved co-expression

While our broad cell type annotations are useful for unifying meta-analysis across heterogeneous primary tissue and organoid data sets, it is also of interest the degree neural organoids are capable of producing primary tissue cell types at a finer resolution, typically through the lens of region-specific cell types. As our approach for quantifying the preservation of co-expression is derived from a genome-wide co-expression network of primary neural tissue, we can also putatively assess preserved co-expression of more specific cell type markers. We investigate preserved co-expression of region-specific cell type markers by utilizing marker genes derived from a morphogen screen in neural organoids that reported the production of extensive neural cell type diversity [73]. As examples of protocol specific trends, we show the directed dorsal forebrain organoid preserves co-expression of telencephalic excitatory neuron markers over markers for mid-brain and thalamic excitatory neurons as well as dopaminergic mid-brain neurons (Fig 4D). Similarly, the directed ventral mid-brain organoid protocol, which reported production of dopaminergic neurons, preserves co-expression of dopaminergic neuron markers over excitatory neuron markers on average (Fig 4D). Extending across all the organoid data sets, we demonstrate preserved co-expression of region-specific cell types exhibit high correlations with the preservation of our broader class-level markers for several glutamatergic and GABAergic cell types (Fig 4E). In summary, our results show that disruption of co-expression at one level of cell type hierarchy captures disruption at finer levels, suggesting a single score for organoid fidelity can capture shared variation. More generally, our quantification for preserved co-expression in organoids can also be applied to the study of region-specific cell types to study variation from the shared baseline.

## Genome-wide preservation of co-expression reveal consistent organoid deficits

In addition to investigating cell type-specific variation for preserving primary tissue co-expression within organoids, our co-expression networks additionally allow genome-wide assessments of preserved co-expression. We extend our analysis via GO terms to quantify preserved primary tissue co-expression within organoids across the whole genome. GO terms with significantly preserved primary tissue co-expression (see Methods) in organoids are mostly related to basic cellular functions like response to DNA damage and protein translation, as well as GO terms related to neurodevelopment (Fig 4F). GO terms that significantly lack preservation of primary tissue co-expression are largely related to angiogenesis or immune function (Fig 4F), concordant with the fact that organoids lack vasculature and an immune system. These results demonstrate quantifications of preserved co-expression can capture known biological deficits in neural organoids.

While GO terms are useful for partitioning the genome into functional units for comparison, our co-expression networks also enable assessments of preserved co-expression for individual genes. As a particular use-case, we search for genes with exceptionally high preserved primary tissue co-expression across primary tissue data sets that also have poor preserved primary tissue co-expression across organoid data sets. We only consider genes that have some measurable expression for the vast majority of data sets (excluding genes with zero expression in more than 5 organoid or primary tissue data sets) and compute the average preserved co-expression AUROC for each gene across the organoid and primary tissue data sets (S8B Fig). The top 10 enriched GO terms for genes with high primary tissue (average AUROC $> = 0.99$) and low organoid (average AUROC $< 0.70$) preserved co-expression are related to extra-cellular matrix (ECM) and vascular characterizations (Fig 4G). The poor conservation of genes related to vasculature can be explained by the absence of vascularization in the vast majority of our organoid data sets. We repeat this analysis using primary tissue co-expression networks derived from only the 6 broad neural cell types, excluding all other cell types including vasculature (S8C Fig). We find that the same ECM-related GO terms are the top ranked terms (by $p$-value) for genes with persistent low preserved co-expression in neural organoids. There are 6 ECM-related genes shared between these 2 experiments: ITGA1 (collagen and laminin receptor), LAMB1 (laminin protein), CD36 (collagen receptor), DISC1 (scaffold protein), PARVB (actin-binding protein), and MMP28 (metalloproteinase). DISC1 is of particular note due to its well-documented association with a wide-range of neuropsychiatric disorders [74], positioning its consistent co-expression dysregulation (DISC1 is expressed similarly to primary tissue and exhibits consistently poor preserved co-expression, S8D Fig) as acutely motivating for further field-wide assessments of neural organoids.

In summary, we interrogate co-expression in organoids at multiple levels, revealing organoids vary in preserving primary tissue co-expression at gene, cell type, and whole genome resolutions through the use of a robust aggregate primary tissue co-expression network. We demonstrate the applicability of our approach for quantifying primary tissue fidelity in organoids against a variety of use-cases, such as comparing across variability in cellular composition, comparing normal and perturbed organoids, and investigating preserved co-expression of individual genes and region-specific cell type markers.

## Temporal variation in organoid preservation of primary tissue co-expression

We score preserved co-expression in organoids using the aggregate primary tissue co-expression network, which by design aims to capture signal robust to temporal variation. To investigate temporal trends in organoid co-expression, we employ a similar approach as when

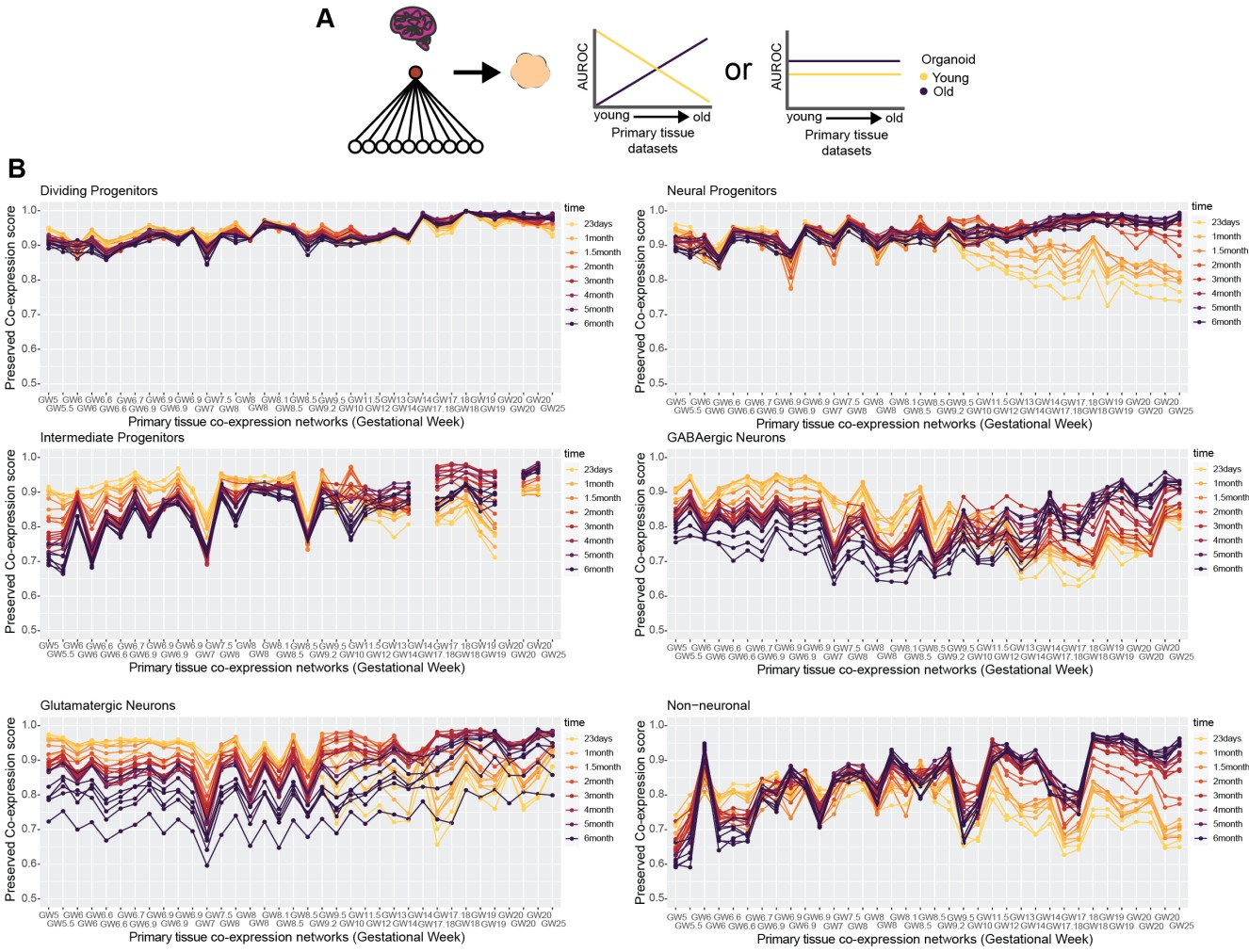

**Fig 5. Neural organoids capture temporal dynamics in primary tissue co-expression. (A)** Schematic showing 2 potential outcomes when comparing the preserved co-expression between primary tissue and organoid data on a temporal axis. There may be a temporal relationship, with younger organoids recapitulating younger primary tissue co-expression over older primary tissue co-expression and vice versa for older organoids, or there may be no temporal relationship. **(B)** Organoid co-expression produces temporal trends in primary tissue co-expression. Line plots showing the preserved co-expression scores computed from individual organoid co-expression networks for cell type markers of individual primary tissue data sets. Primary tissue data sets on the x-axis are ordered from youngest to oldest. Underlying data can be found in the Zenodo repository (doi:10.5281/zenodo.13946248).

predicting organoid cell type annotations in S3 Fig, this time quantifying the preservation of primary tissue co-expression for the top 100 cell type markers per individual primary tissue data set across all organoid time points (Fig 5A and 5B). We uncover a broad temporal shift in the preservation of primary tissue co-expression within organoids across all cell types, with younger organoids (23 days to 1.5 months) as the top performers for mostly first trimester primary tissue co-expression transitioning to older organoids (2 to 6 months) as top performers for mostly second trimester primary tissue co-expression (Fig 5B). This temporal shift appears consistent across the cell types, beginning around GW9-10 (Fig 5B). Our approach in predicting organoid annotations in S3 Fig is based on aggregate marker expression and did not produce temporally variable results, whereas our approach here comparing preserved co-expression of the same marker genes does produce temporally variable results. This indicates that the co-expression relationships of genes rather than their expression levels better capture temporal variation in developing systems.

## Organoids preserve developing brain co-expression over adult brain co-expression

We demonstrate temporal variation in developing brain co-expression relationships is captured by organoids, but only from the single directed dorsal forebrain organoid protocol used in the temporal organoid atlas. In order to extend analysis across all our organoid data sets and assess broad temporal variation in co-expression, we next investigate the preserved co-expression within organoids of both developing and adult brain co-expression relationships.

We construct an aggregate adult co-expression network from a medial temporal gyrus scRNA-seq data set of 155,781 cells [54]. We compare the preserved co-expression scores of organoids for either developing or adult glutamatergic, GABAergic, and non-neuronal cell-types. Organoids almost unanimously preserve developing brain co-expression over adult co-expression (S8E Fig) for all 3 cell types. We extend this analysis genome-wide and place organoids in context between developing and adult data by computing the average preservation of co-expression AUROC across all genes for organoid, developing, and adult co-expression using the annotated primary developing brain tissue network as the reference. The adult co-expression network produces a global preserved developing brain co-expression score of 0.591, indicating very poor performance across the genome in preserving developing co-expression relationships (S8F Fig). Organoids vary substantially in their global preservation of developing brain co-expression with some organoid data sets performing comparably to the adult data. This result is largely influenced by the number of cells present within individual organoid data sets (S8F Fig, corr 0.562, $p$-value $<0.001$), suggesting a cell-sampling limitation for uncovering developing brain co-expression within organoids. However, organoid data sets report more variable global preserved co-expression scores compared to down-sampled developing brain data (S8F Fig), indicating a remaining gap between primary developing brain tissue and organoid data not explained through cell number sampling alone.

We further explore the applicability of our preserved co-expression quantifications for investigating temporal variation through a study that tested the limits of neuronal maturation in organoids. This study generated data from human directed cortical organoids either transplanted or not into developing rat brains to test the limits of maturation organoids can achieve in vitro [67]. We compare the preservation of developing and adult co-expression between these age-matched non-transplanted and transplanted human cortical organoids. We report that for glutamatergic co-expression, non-transplanted organoids preserve developing brain co-expression over adult (non-transplanted; mean developing glut. AUROC: 0.807, mean adult glut. AUROC: 0.672), whereas the transplanted organoids preserve adult co-expression over developing brain (transplanted; mean developing glut. AUROC: 0.766, mean adult glut. AUROC: 0.850, S8G Fig). There was weak preservation of GABAergic co-expression for both developing and adult brain, with slight decreases for both in the transplanted organoids (non-transplanted; mean developing GABA. AUROC: 0.686, mean adult GABA. AUROC: 0.585. Transplanted; mean developing GABA. AUROC: 0.606, mean adult GABA. AUROC: 0.544), as expected of a directed cortical organoid. And for the non-neuronal co-expression, transplantation resulted in slight increases in scores for both developing and adult brain, but the non-transplanted and transplanted organoids had higher preservation of developing non-neuronal co-expression in both cases (non-transplanted: mean developing nonN. AUROC: 0.774, mean adult nonN. AUROC: 0.638. Transplanted; mean developing nonN. AUROC: 0.809, mean adult nonN. AUROC: 0.660). In summary, transplantation produced the strongest effects on glutamatergic co-expression, with non-transplanted organoids preserving developing co-expression over adult and transplanted organoids preserving adult co-expression over developing tissue, in agreement with the authors' original observations of increased

maturation of transplanted organoids. By recapitulating known maturation dynamics in organoid models, we demonstrate the broad applicability of preserved co-expression quantifications for investigating a range of biological phenomenon in neural organoids.

## Variability in organoid co-expression is driven by marker gene expression

We investigate the impact of various technical features in our analysis on our co-expression results by assessing their correlation with our various co-expression scores, specifically focusing on the number of cells, gene detection rate, and percent of mitochondrial mapping reads. Comparing the genome-wide preserved co-expression score (average preserved co-expression AUROC across all genes) per organoid data set to these technical features, we find the size of the data set (number of cells) to have the clearest relation to performance (corr: 0.562, $p$-value: <0.001, S9A Fig), though with top performing data sets ranging from 100s to 10,000s of cells. We also show the genome-wide preserved co-expression score has little relation to the normalization used across organoid studies, with the exception of the Seurat SCTransform normalization that is utilized by a single study which also has the lowest gene detection rate among all the organoid data sets (S9A Fig). An important technical consideration for our analysis is ensuring all data sets have an identical gene namespace for meaningful comparisons of expression data. We fit all data sets to the GO gene universe, dropping gene annotations not in GO and zero-padding missing GO annotations in individual data sets. Excessive zero-padding of genes within our MetaMarker gene sets may artificially lower co-expression module scores or preserved co-expression scores, though we find this relationship to be relatively weak with little impact on score variance (S9B and S9C Fig, $R^2$ for co-expression module scores and zero-padding: 0.0486, 0.0292, 0.0506, 0.0562, 0.0480, 0.158. $R^2$ for preserved co-expression and zero-padding: 0.408, 0.104, 0.132, 0.0763, 0.117, 0.168 for dividing prog., neural prog., intermediate prog., glutamatergic, GABAergic, and non-neuronal cell types, respectively). Comparatively, the correlations between marker set expression and co-expression module scores or preserved co-expression scores are consistently higher (S9B and S9C Fig, range of significant positive ($p$-value <0.001) correlations: 0.245–0.616, $R^2$: 0.0600–0.379 excluding dividing progenitors).

In numerous studies, neural organoids consistently show increased expression of stress-related genes compared to primary tissue [5,12,44,45]. We confirm that finding here and examine its correlation with our preserved co-expression scores. While neural organoids exhibit general increased expression of stress-related genes (glycolysis [45], endoplasmic reticulum-stress [5], and oxidative-stress [5]), there is considerable variability among data sets, with many showing levels comparable to primary tissue (S10A Fig). Stress-related gene expression in neural organoids shows minimal correlation with our preserved co-expression scores at the genome level (S10B Fig), with only slight negative correlations between cell type preserved co-expression and ER-stress expression specifically (S10C Fig). This suggests that ER-related stress pathways may have an increased impact on cellular identity within neural organoids compared to other stress pathways. Additionally, we observe high variability in stress-related gene expression across different organoid protocols (S10D and S10E Fig), indicating different protocols may have distinct impacts on cellular stress.

## Preservation of primary tissue co-expression as a generalizable quality control metric

As a general summary, our approach for quantifying preserved primary tissue co-expression across numerous organoid protocols reveals the axes on which organoids lie for recapitulating primary tissue co-expression relationships at gene, cell-type, and whole-genome resolutions. These assessments provide powerful quality control information, identifying which genes and/

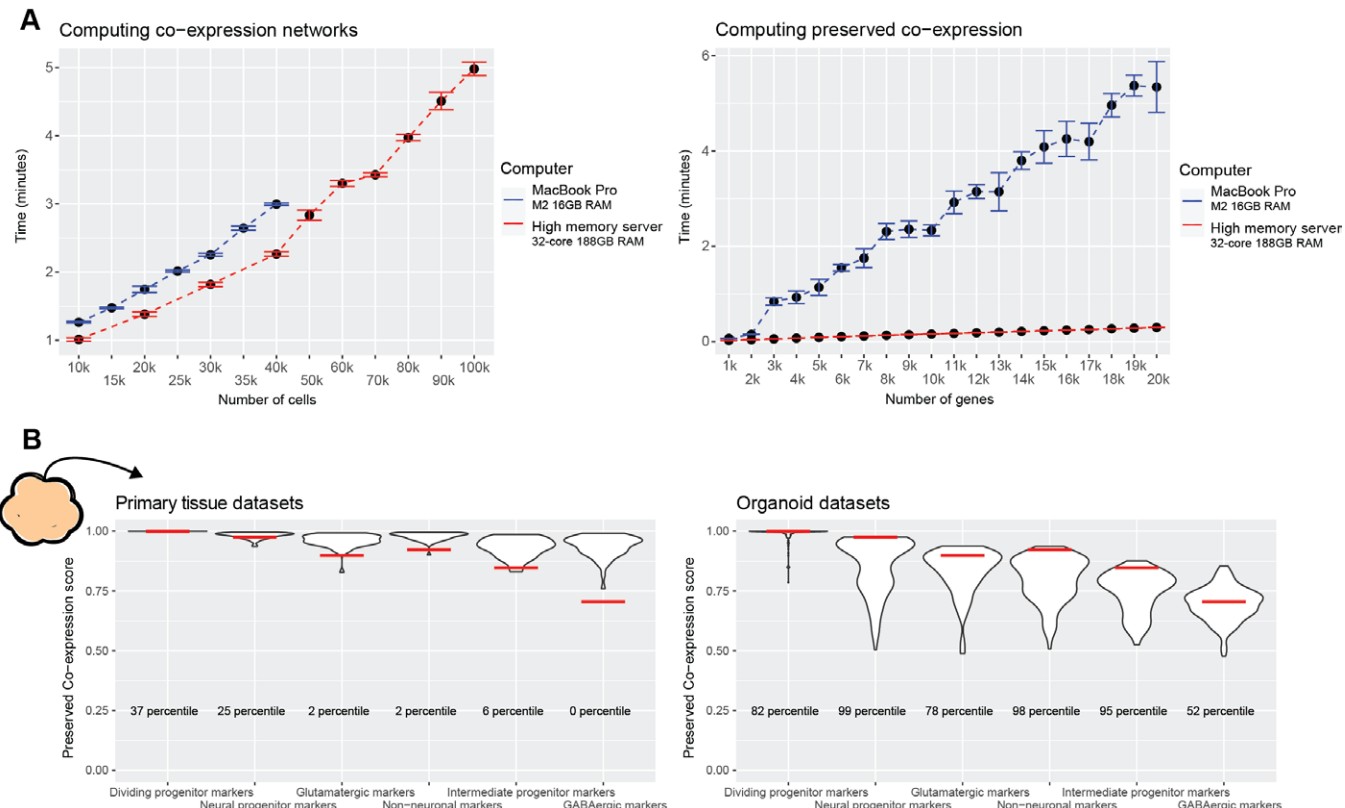

**Fig 6. The preservedCoexp R package enables fast computation of preserved co-expression. (A)** The preservedCoexp R package can compute co-expression networks and genome-wide preservation of co-expression in a few minutes even for low-memory computers. Line plots showing the computational time to either compute co-expression networks or preserved co-expression as the number of cells or genes increases. Points are the mean value from 10 replicates, with error bars depicting ± 1 standard deviation. **(B)** Example plot from the preservedCoexp R package, placing cell type-specific preserved co-expression scores of an example forebrain organoid data set in reference to scores derived from primary tissue data sets or organoid data sets. Red lines denote the percentile of the forebrain organoid cell type scores within either the primary tissue distributions or organoid distributions. Underlying data can be found in the Zenodo repository (doi:10.5281/zenodo.13946248).

or cell types organoids can or cannot currently model on par with primary tissue data. We make our methods accessible through an R package to aid in future organoid studies and protocol development, providing means for rapidly constructing co-expression networks from scRNA-seq data (Fig 6A) as well as querying preserved co-expression of users' data with our aggregate primary tissue brain co-expression network (Fig 6A). Additionally, we make the results of our meta-analysis across primary tissue and organoid data sets available for users to place their data in reference to a field-wide collection (Fig 6B).

## Discussion

Through the use of meta-analytic differential expression and co-expression, we are able to provide cell type-specific measurements of human neural organoids' current capacity to replicate primary tissue biology. We extracted broad cell type markers that define primary brain tissue cell types across a large temporal axis (GW5–25) and across numerous heterogenous brain regions to act as a generalizable primary tissue reference for organoids that also vary temporally and regionally (by protocol). By quantifying intra- and inter-marker set co-expression and the preservation of co-expression across networks, we revealed human neural organoids lie on a spectrum of near-zero to near-identical recapitulation of primary tissue cell type

specific co-expression in comparison to primary tissue data. We made our aggregate primary tissue reference data and methods for measuring preserved co-expression publicly available as an R package to aid in the quality control and protocol development of future human neural organoids.

Prior work comparing primary brain tissue and neural organoid systems demonstrated organoids can produce cell types [11,12] and morphological structures [27,43] similar to primary tissues and are capable of modeling temporal [13,38,40] and regional [3,12,28,29] primary tissue variation. Multiple lines of evidence support these findings such as assessments of cytoarchitecture and cell type proportions [3,11,16,23], whole transcriptome and marker gene expression correlations [10,12], and comparisons of co-expression modules [5,13,17,39]. Our meta-analytic approach is able to quantify these field-wide observations within a generalizable framework, recapitulating that organoids model broad primary tissue biology with our specific approach offering several key advancements for primary tissue/organoid comparisons. First, we derive quantifications of preserved primary tissue co-expression that can be extended from individual genes to the entire genome and, second, we place organoid co-expression in reference to robust meta-analytic primary tissue performance providing a general benchmark for protocol development and quality control across heterogeneous organoid systems. Importantly, we largely avoided direct comparisons of neural organoids across differentiation protocol labels, as naming schemes for protocols are inconsistent across studies and the field as a whole is actively working towards a consensus nomenclature [75], which will require a significant amount of experimental work to fully characterize each protocol. Instead, we grouped data sets empirically through cell type composition to ensure comparisons were made across similar data sets.

In neural organoid systems, a consistent finding is the elevated expression of stress-related genes compared to primary tissue, as confirmed in our analysis. However, there are conflicting conclusions regarding the impact of this increased stress-related expression. Some studies report an impairment of cell type identity among stressed cells [12,44], while others report little effect on core cellular identity programs [38,45]. In our study, we observe a slight negative correlation between cell type-specific preserved co-expression scores and ER stress-related gene expression, with minimal association found for glycolysis and oxidative stress genes. These findings support the notion that cell stress may compromise cellular identity within neural organoids. Importantly, we note significant variability in stress gene expression across organoid protocols, which may contribute to the discordant observations in previous studies. Overall, extensive experimental work is necessary to investigate the potential relationship between cellular stress and cell type identity in in vitro systems further.

While comparisons between primary tissue and organoid systems at a high resolution of cell type annotation are certainly of interest, our results, which focus on broad cell types at the cell-class level, form a critical foundation for these more fine-tuned investigations of organoids. Cell type specification within the brain involves complex spatial and temporal mechanisms [76] to produce the high cellular heterogeneity we observe, with the exact resolution of meaningful cell type annotations still being actively debated and posing a general conceptual challenge within the field of single-cell genomics [77]. We focus here on establishing methods for assessing consistent and accurate production of primary tissue cell types at the class-level within organoids as a critical actionable first step towards increasing primary tissue fidelity across variable organoid differentiation protocols. While we prioritize broad cell type comparisons, we also display the flexibility of our approach by scoring the preserved co-expression of region-specific cell type markers. This demonstrates our quantifications of preserved co-expression are applicable to a variety of cell type annotation resolutions.

One exciting application for the use of neural organoid systems is the study of a wide-range of human neurological diseases using human in vitro models [78,79], which critically depends on the in vivo fidelity of cell types produced in organoids. Neural organoids have been used to model and investigate human disorders of neurodevelopmental [3,80], neuropsychiatric [81–83], and neurodegenerative [58,65,84] nature, as well as infectious diseases [28,85,86]. It is essential that organoid systems model in vivo cell types with extreme fidelity to fully realize the therapeutic potential of human organoids and ensure findings in these in vitro models are not specific to potential artifactual or inaccurate in vitro biology. Our observation of consistent dysregulation of DISC1 among neural organoids is a prime example; DISC1 is associated with a large range of neuropsychiatric disorders [74] and its general dysregulation that we report across neural organoids may obfuscate in vivo and in vitro comparisons. While our results demonstrate that high primary tissue fidelity in organoids is currently methodologically possible, we find this to be the case for a small minority of data sets coupled with a high degree of variability across data sets, indicating a substantial remaining methodological gap. The broad applicability of our meta-analytic approach offers the potential for benchmarking primary tissue fidelity across numerous organoid protocols, aiding in increasing the quality of neural organoids for use in a wide-range of human health-related translational investigations.

## Methods

### Data set download and scRNA-seq preprocessing

Links for all downloaded data (GEO accession numbers, data repositories, etc.) are provided in S1 Table. All scRNA-seq data was processed using the Seurat v4.4.0 R package [87]. Data made available in 10XGenomics format (barcodes.tsv.gz, features.tsv.gz, matrix.mtx.gz) were converted into Seurat objects using the Read10X() and CreateSeuratObject() Seurat functions. Data made available as expression matrices were converted into sparse matrices and then converted into Seurat objects using the CreateSeuratObject() function. Ensembl gene IDs were converted into gene names using the biomaRt v2.20.1 [88] package.

Where metadata was made available, we separated data by batch (Age, Donor, Cell line, etc.) for our final total of 173 organoid and 51 primary tissue data sets (S1 Table). We processed and analyzed each batch independently without integration. We used consistent thresholds for filtering cells across all data sets, keeping cells that had less than 50% of reads mapping to mitochondrial genes and had between 200 and 6,000 detected genes. Several data sets provided annotations for potential doublets; we excluded all cells labeled as doublets when annotations were made available. All data made available with raw expression counts were CPM normalized with NormalizeData(normalization.method = 'RC', scale.factor = 1e6), otherwise normalizations were kept as author supplied.

For primary tissue and organoid data made available with cell type annotations, we provide our mapping between author provided annotations and our broad cell type annotations in S2 Table.

### Primary tissue MetaMarker generation and cross-validation

MetaMarkers were computed using the MetaMarkers v0.0.1 [70] R package, which requires shared cell type and gene annotations across data sets to derive a ranked list of MetaMarkers. Gene markers for individual data sets were first computed using the compute_markers() function on the CPM normalized expression data for our annotated primary tissue data sets (S1 Table). A ranked list of MetaMarkers was then computed using the make_meta_markers() function using all 37 individual annotated primary tissue data set marker lists. Genes are first ranked through their recurrent differential expression (the number of data sets that gene was

called as DE using a threshold of log2 FC $> = 4$ and FDR-adjusted $p$-value $< = 0.05$) and then through the averaged differential expression statistics of each gene across individual data sets. When we take the top 100 markers per individual data set as in Figs 2D, 5, S1A and S3B, we rank markers for each data set by their AUROC statistic as computed with the compute_markers() MetaMarkers function. The top 20 ranked MetaMarkers per cell type are made available in S3 Table and the full lists of top 100 MetaMarkers are made available in the preservedCoexp R package.

For the cross-validation of our primary tissue MetaMarkers, we excluded a single annotated primary tissue data set, computed MetaMarkers from the remaining 36 annotated primary tissue data sets, and then used those MetaMarkers to predict the cell type annotations of the left-out data set. We construct an aggregate expression predictor to quantify the predictive strength a list of genes has, in this case our MetaMarker lists, in predicting cell type annotations. Taking any arbitrary number of genes (10, 20, 50, 100, 250, or 500 MetaMarkers), we sum the expression counts for those genes within each cell and then rank all cells by this aggregate expression vector. We compute an AUROC using this ranking and the cell type annotations for a particular cell type through the Mann–Whitney U test. Formally:

$$AUROC = \frac{U}{n_0 * n_1}$$

where U is the Mann–Whitney U test statistic, $n_0$ is the number of positives (cells with a given cell-type annotation), and $n_1$ is the number of negatives (cells without that cell type annotation).

$$U = R_0 - \frac{n_0(n_0 + 1)}{2}$$

where $R_0$ is the sum of the positive ranks.

As an example, if there are 10 genes that are perfect glutamatergic markers (only glutamatergic cells express these genes), then ranking cells by the summed expression of these genes will place all glutamatergic cells (positives) in front of all other cells (negatives), producing an AUROC of 1. The violin plots in S1B Fig and in Fig 2E visualize our aggregate expression approach, where data points per cell type are the aggregated expression counts for the given top 100 MetaMarkers across all cells per data set (S1B Fig) or aggregated across all data sets (Fig 2E). We also compared the aggregate expression of the Neural Progenitor MetaMarkers across author provided cell type annotations included in our broad Non-neuronal annotation, revealing the off-target expression of Neural Progenitor MetaMarkers is specific to annotated astrocytes (S1B Fig).

For S1A Fig, we took the top 100 cell type markers per individual primary tissue data set (x-axis) and used those genes to predict cell type annotations as described above for all other annotated primary tissue data sets, reported as the AUROC boxplot distributions. The Meta-Marker distribution was computed using a leave-one-out approach as described above. We ranked the individual primary tissue data sets by their median AUROC performance per cell type to derive the distributions of ranks presented in Fig 2D, excluding the dividing progenitor data as performance was highly consistent across all primary tissue data sets.

## Cross-regional primary tissue MetaMarker expression

We investigated the aggregate expression of our top 100 MetaMarkers per cell type across annotated brain regions separately for the annotated first trimester and second trimester primary tissue atlases due to differing regional annotations. MetaMarkers were computed with a

leave-one-out approach as described above using all 37 of the annotated primary tissue data sets. For the heatmaps in S2 Fig, rows represent the annotated cells present within the given data set, columns represent the aggregated expression for the top 100 given cell type MetaMarkers and each annotated region present. We average the aggregated expression for each cell type per region and then normalize each region (column) by the maximum average expression value across the cell types. A value of 1 indicates that cell type is the one maximally expressing the given MetaMarker set for that brain region. The heatmaps are ordered by cell type and region and are not clustered.

## Organoid PCA

PCA analysis was performed using the Seurat function RunPCA() with the top 2,000 variable features, determined using the Seurat function FindVariableFeatures(selection.method = "vst", nfeatures = 2,000). For each organoid data set, we took the eigenvector for the first principal component, computed the absolute value, and then divided by the maximum value to compute a normalized vector between 0 and 1. We visualized the normalized eigenvectors for each organoid data set in S3C Fig, keeping primary tissue MetaMarker genes that were detected in the top 2,000 variable genes of at least 10 organoid data sets. Genes missing from any given data set's top 2,000 variable genes were given a value of 0. The heatmap was produced using the ComplexHeatmap v2.12.1 [89] package and was hierarchically clustered using the ward.D2 method for both rows and columns.

## Generating co-expression networks from scRNA-seq data

To generate a shared gene annotation space across all data sets, we fit each data set to the GO gene universe before computing co-expression matrices. Using human GO annotations (sourced 2023-07-27 using the org.Hs.eg.db v3.18.0 [90] and AnnotationDbi v1.64.1 [91] R packages), we excluded gene expression from a data set if the gene annotation was not present in GO and we zero-padded missing GO genes for each data set.

We compute a gene-by-gene co-expression matrix per data set using the Spearman correlation coefficient computed across all cells in a given data set. We then rank the correlation coefficients in the gene-by-gene matrix and divide by the maximum rank to obtain a rank-standardized co-expression matrix. All results reported using individual data set co-expression networks (Figs 3D, 3E, 4B, 5, 6 and S4–S9) were obtained using the rank-standardized co-expression networks.

We compute the aggregated co-expression networks by taking the average of the rank standardized co-expression networks for each gene-gene index and then rank-standardizing the averaged network.

## Co-expression module learning analysis

EGAD v1.30.0 [71] is a machine learning framework that quantifies the strength of co-expression within an arbitrary gene-set compared to the rest of the genome with an AUROC quantification (Fig 3C). We compute co-expression module AUROCs for all GO gene-sets (between 10 and 1,000 genes per GO term) and our top 100 primary tissue MetaMarker gene sets for each individual primary tissue and organoid co-expression network as well as the aggregated annotated, unannotated and organoid networks. For all co-expression analyses using the top 100 MetaMarkers per cell type, we ignore duplicated markers across cell types. We employ a leave-one-out approach for the annotated primary tissue co-expression networks, learning MetaMarkers from 36 of the annotated data sets and computing co-expression module AUROCs for these MetaMarkers in the left-out data set's co-expression network. We compute co-

expression module AUROCs using the EGAD run_GBA() function with default parameters. In Fig 3D, the "All GO terms" distributions report the average co-expression module AUROC across all GO terms for each individual network.

## MetaMarker inter-marker set co-expression

To score the inter-marker set co-expression of a given gene in a given MetaMarker gene set, we compute the mean co-expression between that gene and all MetaMarkers not in that gene's MetaMarker gene set, divided by the mean co-expression of all MetaMarkers within that gene's MetaMarker gene set. A value less than 1 indicates that gene's co-expression within its MetaMarker gene set is stronger than without, and vice versa for a value greater than 1. We take the average of these ratios across genes within each MetaMarker gene set to summarize at the level of individual networks, with Fig 3E and S5B Fig reporting the average ratios. We center the distributions around the scores of the aggregated unannotated primary tissue network in Fig 3E, with the raw ratios in S5B Fig.

## Organoid cell type composition

We compute organoid cell type composition by annotating organoid cells using the default MetaMarker cell type annotation approach. MetaMarker cell type annotation takes a set of cell type markers, gives each cell a score for each cell type marker set as the average expression of that marker set within the cell (MetaMarker function score_cells()), and then divides all scores by the total average marker set expression to compute marker enrichment (MetaMarker function compute_marker_enrichment). The marker set with the highest enrichment score per cell is used as the predicted cell type annotation for that cell (MetaMarker function assign_cells()). The final cell type percentage we report for each data set is the number of cells annotated for a given cell type annotation divided by the total number of cells within that data set. We use the top 15 primary tissue MetaMarkers, excluding duplicate markers, as the marker sets to predict cell type annotations in organoid data sets, using all MetaMarker package functions with default parameters. We compare the MetaMarker predicted cell type annotations to the author provided cell type annotations in the temporal directed dorsal forebrain organoid data sets (S4A Fig), which we also used to generate the confusion matrix in S4B Fig. S4C Fig depicts the MetaMarker assigned cell type percentages for all the organoid data sets, with columns hierarchically clustered using the ward.D2 algorithm. We note that data sets that used the SCTransform normalization and the default Seurat log normalization with UMI regression generally failed the MetaMarker cell type annotation prediction, with large portions of cells being unable to be assigned a cell type through enriched marker set expression. This is likely due to incompatibilities between the normalization used and the MetaMarker approach of averaging expression values. We exclude these data sets when comparing cell type composition and our co-expression scores in S4D and S5C Figs. When grouping organoid data sets by cell type composition, we bin the data sets using increasing intervals of 10 percentage points per cell type.

## Preservation of co-expression

To compute our preservation of co-expression AUROC, we take the top 10 co-expressed partners for gene A in a reference co-expression network as our positive gene annotations. In a test co-expression network, we rank all genes through their co-expression with gene A and compute an AUROC using this ranking and the positive annotations derived from the reference network. If gene A in the test network has the exact same top 10 co-expressed partners as in the reference network, that would result in an AUROC of 1. To summarize a given gene set's preserved co-expression, we take the average preserved co-expression AUROC across all genes

in that gene set as the preservation of co-expression score for that gene set. For all co-expression analyses using the top 100 MetaMarkers per cell-type, we ignore duplicated markers across cell types. We use the aggregated annotated primary tissue co-expression matrix as our reference network.

The preserved co-expression scores for the annotated primary tissue data in Fig 4B were computed with a leave-one-out approach. MetaMarkers and an aggregated co-expression network were computed from 36 of the annotated primary tissue data sets and then preserved co-expression scores were computed using the co-expression network of the left-out annotated primary tissue data set.

### Preservation of co-expression across individual primary tissue brain regions

We construct co-expression networks for individual annotated brain regions using annotations provided by the original authors in a similar leave-one-out manner as described above, generating an aggregated co-expression network from the remaining annotated primary tissue data sets. The number of cells per sampled brain region as well as the preserved co-expression scores for our 6 broad cell types are presented in S6 and S7 Figs.

### Preservation of region-specific cell types

To define markers for region-specific cell types, we utilize the differential expression (DE) statistics computed from a study that performed a morphogen screen in neural organoids and reported extensive neural cell type diversity [73]. For each cell type, we rank genes by their adjusted DE $p$-value and take the top 10 genes per cell type to compute preserved co-expression scores. When comparing against our MetaMarker gene sets in Fig 4E, we ensure no overlap in the top 10 cell type and top 100 MetaMarker gene sets.

### Preservation of GO term co-expression

We compute $p$-values for the preservation of co-expression of GO terms using a mean sample error approach. Using the aggregated annotated primary tissue co-expression network as the reference and the aggregated organoid network as the test network, we first compute the preserved co-expression AUROCs for all individual genes, taking the mean and standard deviation value as the population mean and population standard deviation. For any given GO term, we first compute the preserved co-expression score for the term (the average of the preserved co-expression AUROCs for the genes in the term) and then compute the sample error for that score with:

$$SE = \frac{SD_{pop}}{\sqrt{n_{GO}}}$$

where $SD_{pop}$ is the population standard deviation and $n_{GO}$ is the number of genes in the GO term. We then compute a z-score through:

$$Z_{GO} = \frac{mu_{GO} - mu_{pop}}{SE}$$

where $mu_{go}$ is the preserved co-expression score for the GO term and $mu_{pop}$ is the population mean preserved co-expression AUROC. We compute left-sided $p$-values using the standard normal distribution:

$$p_L = P(X \leq Z_{GO})$$

where X is a normal distribution with mean = 0 and standard deviation = 1. We use the R function pnorm($Z_{GO}$) to compute this *p*-value.

We then compute the right-sided *p*-value as:

$$p_R = 1 - p_L$$

We adjust *p*-values using the R function p.adjust(method = 'BH'). We filter for GO terms that have between 20 and 250 genes per term and use a threshold of FDR-corrected *p*-value < = 0.0001 to call significance. Significant left-sided *p*-values are interpreted as GO terms with significantly smaller preserved co-expression scores (significantly not preserved) than expected through sampling error and right-sided *p*-values are interpreted as GO terms with significantly larger preserved co-expression scores (significantly preserved) than expected through sampling error. We use the R package rrvgo to visualize the significant GO terms in Fig 4F.

### Computing correlation significance

We employ a permutation test to compute *p*-values for any given correlation coefficient. We permute data pairs and compute a correlation coefficient, repeating for 10,000 random permutations to generate a distribution of correlation coefficients under the null hypothesis of independence. We calculate a two-sided *p*-value for the original correlation coefficient as the number of permuted correlation coefficients whose absolute value is greater than or equal to the absolute value of the original correlation coefficient, divided by 10,000. We adjust *p*-values using the R function p.adjust(method = 'BH') and use a FDR-corrected *p*-value threshold of < = 0.05 to call significance.

### Comparing co-expression of normal versus perturbed organoids

For both the co-expression module AUROCs and the preserved co-expression scores of normal and perturbed organoids, we test for significant differences per cell type using the Mann–Whitney U test, adjusting *p*-values with the R function p.adjust(method = 'BH') and using a FDR-corrected *p*-value threshold of < = 0.05 to call significance.

### Organoid temporal analysis

The organoid temporal analysis for both predicting organoid annotations with primary tissue markers (S3B Fig) and scoring the preserved co-expression of organoid co-expression using primary tissue networks as reference (Fig 5) were performed for all pair-wise combinations of the 37 annotated primary tissue data sets and the 26 temporally annotated directed dorsal forebrain organoid data sets. We excluded the GW7-28 annotated primary tissue data set from the temporal preserved co-expression analysis (Fig 5) due to the wide temporal range sampled. For predicting organoid annotations with primary tissue markers (S3B Fig), we used the top 100 markers per primary tissue data set to construct aggregate expression predictors in the organoid data sets as described above. The MetaMarkers performance was calculated using MetaMarkers derived from all 37 annotated primary tissue data sets. For scoring preserved co-expression, individual primary tissue networks were used as the reference with individual organoid networks as the test networks. We computed the preserved co-expression scores of the top 100 primary tissue cell type markers per individual primary data set for each individual organoid network.

### Stress-related gene expression

We use stress-related gene sets as defined by prior studies [5,45] comparing neural organoids and primary tissue, with all genes provided in S4 Table. Specifically, the glycolysis genes are

the Canonical Glycolysis GO term (GO:0061621) and the ER-stress (organoid_human_ER_-stress_module) and Oxidative stress (organoid_human_oxidative_stress_module) genes were defined through module analysis in [5], as provided in their S3 Table. Only data sets that utilized CPM normalization were used in S10 Fig.

## GO enrichment analysis

We compute enrichment for GO terms using Fisher's exact test as implemented through the hypergeometric test. We compute raw $p$-values for GO terms with between 10 and 1,000 genes and compute FDR-adjusted $p$-values using p.adjust(method = 'BH'). We only consider GO sets with between 20 and 500 when choosing the top 10 GO sets in Fig 4G, ranked by FDR-adjusted $p$-value.

## R and R packages

All analysis was carried out in R v4.4.1. Colors with selected using the MetBrewer v0.2.0 R library. Plots were generated using ggplot2 v3.5.1 [92]. Spearman correlation matrices for co-expression networks were computed using a python v3.6.8 script, implemented in R with the reticulate v1.38 R package, as well as using functions from the matrixStats v1.3.0 R library.

## Supporting information

**S1 Fig. MetaMarkers as temporally robust primary tissue cell type markers. (A)** Boxplots of AUROCs for predicting cell type annotations across all primary tissue data sets using the top 100 marker genes per individual primary tissue data set compared to MetaMarkers (red). Data sets are ordered by their median performance, providing the rank distributions in Fig 2D. **(B)** Distributions of averaged gene expression for the top 100 MetaMarkers across all annotated primary tissue data sets with leave-one-out cross-validation. Fig 2E is the aggregate over these individual data set distributions. Inset displays the average neural progenitor MetaMarker expression for neural progenitor, astrocyte, and all non-astrocyte non-neuronal cells. Underlying data can be found in the Zenodo repository (doi:10.5281/zenodo.13946248). (PNG)

**S2 Fig. MetaMarkers as regionally robust primary tissue cell type markers. (A)** Heatmaps of maximum normalized average MetaMarker expression for cell types and brain regions of the first trimester annotated primary tissue atlas. Cell types comprise the rows with MetaMarker gene expression for cells from each annotated brain region comprising the columns. Data is maximum normalized per region/column. **(B)** Heatmaps of maximum normalized average MetaMarker expression for cell types and brain regions of the second trimester annotated primary tissue atlas. Cell types comprise the rows with MetaMarker gene expression for cells from each annotated brain region comprising the columns. Data is maximum normalized per region/column. Underlying data can be found in the Zenodo repository (doi:10.5281/zenodo.13946248). (PNG)

**S3 Fig. Primary tissue MetaMarkers consistently predict organoid cell types across time points. (A)** Schematic showing 2 potential outcomes when comparing cell type marker expression between primary tissue and organoid data on a temporal axis. There may be a temporal relationship, with younger organoids recapitulating younger primary tissue marker expression over older primary tissue marker expression and vice versa for older organoids, or there may be no temporal relationship. **(B)** Line plots showing the cell type prediction AUROCs using top 100 markers from individual primary tissue data sets for all organoid time points. Primary

tissue data sets on the x-axis are ordered from youngest to oldest. **(C)** Heatmap of min-max normalized eigenvalues for primary tissue MetaMarkers within the first principal component of each organoid data set. **(D)** MetaMarker and non-marker gene set distributions of normalized PC1 eigenvalues across all organoid data sets (left plot). The right plot depicts the fraction of data sets each MetaMarker gene (the average for each MetaMarker gene set is reported) is called as "heavily weighted" in PC1 for a given normalized eigenvalue threshold (x-axis). Underlying data can be found in the Zenodo repository (doi:10.5281/zenodo.13946248). (PNG)

**S4 Fig. Intra-marker set MetaMarker co-expression is weakly related to cell type composition. (A)** Scatter plots comparing the cell type annotation percentage of individual organoid data sets from the annotated directed dorsal forebrain temporal data sets, with author-provided annotations on the x-axis and annotations determined by MetaMarker expression on the y-axis. **(B)** Confusion matrix for the results in A, comparing the MetaMarker predicted annotations to the author provided annotations. **(C)** Heatmap displaying the predicted cell type percentages (rows) of all the organoid data sets (columns), hierarchically clustered by the organoid protocol. The plot to the left of the heatmap depicts the distributions of unassigned cell percentages per data set, where large percentages of unassigned cells are dependent on the expression normalization used in individual studies. Studies with those normalizations were excluded from S4D and S5C Figs. **(D)** Boxplot distributions comparing the predicted cell type percentage (top boxplot plot per cell type, binned in intervals of 10-percentage points) to the co-expression module score (bottom boxplot plot per cell type) for all neural organoid data sets. The x-axes are the same for the top and bottom sets of boxplots per cell type. Underlying data can be found in the Zenodo repository (doi:10.5281/zenodo.13946248). (PNG)

**S5 Fig. Preservation of MetaMarker set co-expression is weakly related to cell type composition. (A)** Boxplots comparing the co-expression module scores (top row) for the neural lineage (neural progenitor, intermediate progenitor, glutamatergic, GABAergic, and non-neuronal MetaMarkers) or microglia/immune MetaMarkers between the neural organoid and non-neural organoid data sets. Bottom row of boxplots depicts the preserved co-expression scores. **(B)** Boxplots depicting the raw inter-marker set co-expression ratios across MetaMarker gene sets for the unannotated primary tissue and neural organoid data sets, standardized ratios are in Fig 3E. Special characters denote the scores of the aggregate co-expression networks. **(C)** Boxplot distributions comparing the predicted cell type percentage (top boxplot plot per cell type, binned in intervals of 10-percentage points) to the preserved co-expression score (bottom boxplot plot per cell type) for all neural organoid data sets. The x-axes are the same for the top and bottom sets of boxplots per cell type. Underlying data can be found in the Zenodo repository (doi:10.5281/zenodo.13946248). (PNG)

**S6 Fig. Primary tissue variability in cross-regional preservation of co-expression is related to dataset size. (A)** Violin and dotplots displaying the number of sampled cells per annotated brain region across the cross-regional first and second trimester primary tissue data sets. **(B)** Violin and dotplots displaying the preserved co-expression scores of the top 100 MetaMarkers for each of our 6 cell type annotations, separated by brain region. Underlying data can be found in the Zenodo repository (doi:10.5281/zenodo.13946248). (PNG)

**S7 Fig. Organoids maintain a deficit in preserved primary tissue co-expression when controlling for data set size and gene detection rates.** (A) Boxplots and dotplots comparing the

preserved co-expression scores for the top 100 dividing progenitor MetaMarkers for all neural organoid data sets (gray) and the region-specific primary tissue data sets (red) from S6 Fig. All data sets are binned on the x-axis by the number of cells present (left-most plot) or the median number of detected genes (right-most plot) per data set. (B) Same as A, but for the top 100 neural progenitor MetaMarkers. (C) Same as A, but for the top 100 intermediate progenitor MetaMarkers. (D) Same as A, but for the top 100 glutamatergic MetaMarkers. (E) Same as A, but for the top 100 GABAergic MetaMarkers. (F) Same as A, but for the top 100 non-neuronal MetaMarkers. Underlying data can be found in the Zenodo repository (doi:10.5281/zenodo.13946248). (PNG)

**S8 Fig. Neural organoids preserve co-expression of developing neural tissue over adult neural tissue. (A)** Boxplots comparing either the co-expression module scores or preserved co-expression scores by cell type between normal and treated organoids. **(B)** Scatter plot showing the average preserved developing brain co-expression AUROC of individual genes, comparing the average across developing brain networks (x-axis) against the average across organoid networks (y-axis). The points colored in red are genes with developing brain scores $\geq 0.99$ and organoid scores $< 0.70$. **(C)** Same as in B, only using primary tissue co-expression networks derived using only the 6 broad annotated cell types. Points in red are genes with developing brain scores $\geq 0.90$ and organoid scores $< 0.70$. The bar plot shows the top 10 GO terms determined by *p*-value for GO set enrichment of the genes in red. **(D)** Scatter plot comparing the expression of DISC1 (log10(CPM+1)) to the preserved co-expression score of DISC1 (computed using primary tissue co-expression networks derived from just the neural lineage cell types) for all primary tissue and organoid data sets that used CPM normalization. The boxplots compare DISC1 expression (left) and DISC1 preserved co-expression (right) between primary tissue and neural organoids. **(E)** Scatterplots showing the preserved co-expression scores of either the top 100 developing brain MetaMarkers (x-axis) or the top 100 adult MetaMarkers (y-axis). **(F)** Distributions of average preserved developing brain co-expression AUROCs across all genes for organoid and developing brain networks. The redline shows the performance of the adult co-expression network. The scatterplot plots the data in the histogram (y-axis) against the number of cells in each organoid data set (x-axis). The blue line shows performance for a cell down-sampled developing brain data set, with points representing the average performance over 10 random samples and the error bars showing ± 1 standard deviation. **(G)** Transplanted organoids preserve adult co-expression over developing brain co-expression. Points represent the log2-fold change over the mean performance of the non-transplanted organoids for preserved co-expression scores. Underlying data can be found in the Zenodo repository (doi:10.5281/zenodo.13946248). (PNG)

**S9 Fig. Strength of MetaMarker co-expression in organoids is related to expression levels. (A)** Scatter plots comparing either the cell number, the median number of detected genes, or the median percentage of mitochondrial mapping genes (x-axis) of each organoid scRNA-seq data set to the average preserved co-expression AUROC across all genes (y-axis). The boxplots display the global preserved co-expression score across the RNA-seq normalizations used among the organoid data sets. **(B)** Scatterplots of either the zero-padded ratio (top row) or marker set expression (bottom row) against the co-expression module scores for each cell type across the organoid data sets. **(C)** Scatterplots of either the zero-padded ratio (top row) or marker set expression (bottom row) against the preserved co-expression scores for each cell type across the organoid data sets. Underlying data can be found in the Zenodo repository (doi:10.5281/zenodo.13946248). (PNG)

**S10 Fig. ER-related stress gene expression is lightly related to preserved co-expression within neural organoids. (A)** Histogram depicting the log2 fold change (FC) of the 76 stress-related genes as defined in S4 Table, including only genes present within all data set gene annotations. The boxplots compare the expression distributions of the genes with at least a mean expression of 50 CPMs across all data sets and defined as either elevated in expression for primary tissue data sets (log2(FC) $< = -0.25$, left set of boxplots) or elevated in expression for neural organoids (log2(FC) $> = 0.25$, right set of boxplots). Expression values are the z-scored CPM values for each gene. Only neural organoid and primary tissue data sets that used CPM normalization were used for this analysis. **(B)** Scatter plots comparing the global preserved co-expression score of all neural organoid datasets against their mean expression (CPM) of either the glycolysis, ER-stress, or oxidative stress genes. **(C)** Heatmap depicting the Spearman correlations between the mean expression of stress-related gene sets (columns) and the cell type-specific preserved co-expression scores (rows) across the neural organoid data sets that used CPM normalization. **(D)** Boxplots depicting the distributions of mean expression (CPM) of either the glycolysis, ER-stress, or oxidative stress genes across the neural organoid protocols/data sets and primary tissue data sets that used CPM normalization. **(E)** Heatmap depicting the Spearman correlations in mean expression (CPM) of the glycolysis, ER-stress, and oxidative stress genes across the neural organoid data sets that used CPM normalization. Underlying data can be found in the Zenodo repository (doi:10.5281/zenodo. 13946248).
(PNG)

**S1 Table. Table containing the study origin and download links for all primary tissue and organoid scRNA-seq data sets.** The batch variable column details the meta-data used in determining batch. The Author described protocol column contains protocol descriptions as utilized in each data set's original publication and the Protocol classification/region column contains protocol descriptions as we classify them in this study. We also provide the age, condition, and cell-line annotations when made available for all studies, as well as various technical and quality control-related factors for all data sets.
(XLSX)

**S2 Table. Table containing our mapping between authors provided annotations (Author annotations column) and our broad cell type annotations (Class annotations column).**
(XLSX)

**S3 Table. Table containing the top 20 ranked MetaMarkers for the dividing progenitor, neural progenitor, intermediate progenitor, glutamatergic, GABAergic, non-neuronal, and microglia/immune cell type annotations.**
(XLSX)

**S4 Table. Table containing cellular stress gene sets.**
(XLSX)

## Acknowledgments

We thank Tomasz Nowakowski and members of the Gillis lab for helpful comments on the manuscript.

## Author Contributions

**Conceptualization:** Jonathan M. Werner, Jesse Gillis.

**Data curation:** Jonathan M. Werner.

**Formal analysis:** Jonathan M. Werner, Jesse Gillis.

**Funding acquisition:** Jesse Gillis.

**Investigation:** Jonathan M. Werner, Jesse Gillis.

**Methodology:** Jonathan M. Werner, Jesse Gillis.

**Project administration:** Jesse Gillis.

**Resources:** Jonathan M. Werner, Jesse Gillis.

**Software:** Jonathan M. Werner.

**Supervision:** Jesse Gillis.

**Validation:** Jonathan M. Werner, Jesse Gillis.

**Visualization:** Jonathan M. Werner.

**Writing – original draft:** Jonathan M. Werner, Jesse Gillis.

**Writing – review & editing:** Jonathan M. Werner, Jesse Gillis.

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
