## [Editor Report · Decision Letter 0]

7 Dec 2023

Dear Dr Gillis, 

Thank you for submitting your manuscript entitled "Preservation of co-expression defines the primary tissue fidelity of human neural organoids" for consideration as a Research Article by PLOS Biology. Please accept my sincere apologies for the long delay in getting back to you as we consulted with an academic editor about your submission.

Your manuscript has now been evaluated by the PLOS Biology editorial staff and I am writing to let you know that we would like to send your submission out for external peer review.

Once your full submission is complete, your paper will undergo a series of checks in preparation for peer review. After your manuscript has passed the checks it will be sent out for review. To provide the metadata for your submission, please Login to Editorial Manager (https://www.editorialmanager.com/pbiology) within two working days, i.e. by Dec 09 2023 11:59PM.

Kind regards,

Richard

Richard Hodge, PhD

rhodge@plos.org

PLOS

---

## [Decision Letter · Decision Letter 1]

13 Feb 2024

Dear Dr Gillis,

Thank you for your patience while your manuscript "Preservation of co-expression defines the primary tissue fidelity of human neural organoids" was peer-reviewed at PLOS Biology. Please accept my sincere apologies for the long delays that you have experienced during the peer review process. Your manuscript has now been evaluated by the PLOS Biology editors, an Academic Editor with relevant expertise, and by four independent reviewers. 

In light of the reviews, which you will find at the end of this email, we would like to invite you to revise the work to thoroughly address the reviewers' reports.

As you will see, the reviewers are generally positive and think that the meta-analysis is useful and will provide a significant contribution to the organoid field. However, Reviewer #2 is more negative and thinks that the organization of the datasets is causing batch-effects in the analyses and artificial conclusions. After discussions with the Academic Editor, we ask that a revised version addresses these concerns around the preprocessing step for the meta-analysis and provides clarifications about the methods compared and how they perform.

Given the extent of revision needed, we cannot make a decision about publication until we have seen the revised manuscript and your response to the reviewers' comments. Your revised manuscript is likely to be sent for further evaluation by all or a subset of the reviewers.

**IMPORTANT - SUBMITTING YOUR REVISION**

*Re-submission Checklist*

*Published Peer Review*

*PLOS Data Policy*

*Blot and Gel Data Policy*

Kind regards,

Richard

Richard Hodge, PhD

rhodge@plos.org

REVIEWS:

Reviewer #1: Werner and Gillis perform a meta-analysis of neural organoid gene expression to identify how consistent the signature is. This is an important endeavor - organoid model systems are a proxy for a biological system (ie, tissue or tissue-tissue interaction) and assumed to reflect it with sufficient fidelity. Gene expression, specifically transcriptional modules, are probably the best way to answer this since they have been shown to be fairly robust in terms of multiple data points changing simultaneously, giving additional statistical power to estimations. Gillis uses MetaMarker, which he published previously, as part of this assessment and shows that relatively few genes can robustly predict cell type.

Their sample size is sufficiently powered - millions of cells across 180 scRNAseq datasets total. They compare primary tissue, using different timepoints and brain regions to identify a core, robust signal, with organoids. They also show organoids vary in preserving primary tissue co-expression at the gene and cell-type level, and that their approach can quantify primary tissue fidelity in organoids. They flesh out important caveats to the approach and discuss limitations. 

The manuscript is quite polished and I don't have any significant concerns. This type of assessment is valuable for the scRNAseq community and should be published. Finally, they provide an R package for the community to compare their own samples, which is very valuable.

Minor concerns

The figures are blurry. It may be due to the conversion to PDF, but it should be fixed prior to publication.

Reviewer #2: Werner and Gillis collected publicly available single-cell RNA-seq data of human fetal/primary brains and neural organoids from NCBI GEO and performed meta-analysis of in vivo and in vitro brain transcriptomics. Surprisingly, they collected over 4 million cells from 181 datasets, which are invaluable reference data resource for comparison between in vivo and in vitro brain system, and across different protocol types of organoids. Focusing on gene co-expression is also interesting topic, since co-expression was not deeply addressed in previous meta-analyses of scRNA-seq in neural organoids (Bhadiri et al., Tanaka et al.). However, I feel that the collected datasets were not well organized, and concerns that their conclusion is artifact derived from the unorganized datasets. My major and minor concerns are shown as follow:

Major concerns:

* It is unclear how authors defined 12 protocol types. What is difference between "neural_induced_blood_vessel" and "vascularized_cortical"? What is difference between "dorsal_patterned_forebrain" and "cortical"? Most of cortical organoids are guided by dorsal patterning. Bhadiri et al. Nature 2020 clearly defined these cortical organoid protocols as least directed, directed, and most directed. If authors purpose to assess biological fidelity across different protocols, authors should carefully digest each organoid protocol, and systematically classify them: guided or unguided, directed region, stringency of patterning.

* In Fig. 1B, "vascularized_cortical" has 1 dataset. However, this is not correct. Shi et al. PLoS Biol 2020 provided total 12 datasets (6 vascularized and 6 non-vascularized cortical organoids). I think that authors counted some datasets by series accession number (GSEXXXX) in certain datasets (e.g. Shi et al., Xiang et al.), but other dataset by sample accession number (GSMXXXX) (e.g. Velasco et al., Quadrato et al.). The number of dataset should be counted by biological replicates.

* Xiang et al. also provided two different types of organoids (cortical and MGE). Authors should separate this dataset into two protocols.

* Is Femando et al dataset for retina-cortical organoid? According to NCBI GEO, this is a cortical organoid.

* Why organoid ages are not provided in certain protocols (e.g. Pollen et al., Xiang et al.)? These datasets also provided organoid age in NCBI GEO or in the article.

* Authors collected single-cell transcriptome data, which were derived from different platforms (e.g. 10X Genomics Chromium, STRT-seq, Fluidigm C1). Since the number of detected genes varies with single-cell platform, it is possible that the co-expression patterns are also affected by the platform difference. 

* Improve the summary table of dataset (Supp Table 1). Please add more valuable information, such as single-cell platform, organoid type (guided or unguided), and organoid/brain age, donor ID/cell line. In addition, certain datasets lack GSM accession number. For example, Shi et al. performed scRNA-seq to vascularized and non-vascularized organoids with different ages (d65, d100). It is very important which accession number corresponding to vascularized organoid or normal organoid.

* This unorganized dataset may cause unfair batch effect normalization across datasets, and possibly affect gene co-expression patterns.

* Please provide more detail of their cell-type annotations. How to determine 6 cell types? "Non-neuronal" contains various cell types, including astrocyte, oligodendrocyte, and microglia, which express clearly distinguishable markers (e.g. AQP4, MBP, AIF1). In meta-analysis, I strongly recommend uniformly determining cell types in each dataset by user-defined rules (e.g. markers) or automatic annotation software.

* In public data, some low-quality libraries were deposited (in particular, scRNA-seq library for primary/fetal tissues). Therefore, these low-quality libraries should be removed before co-expression analysis. Although authors performed QC with %mitochondria and detected genes, what percentage of cells were excluded by these criteria?

* Trevino et al. dataset was classified as unannotated dataset, but cell type annotation is provided from NCBI GEO ("multiome_cluster_names.txt.gz")

* At line 144-147, authors described that they identified sets of marker genes to define cell types across brain regions. However, they did not provide any results related to brain regions (Fig. 2 was performed across timepoint). Although Bhadiri et al. and Braun et al. performed scRNA-seq in multiple brain regions, whether authors analyzed them in each brain region separately or merged all brain regions?

* It is interesting that a small number of meta-analytically derived primary tissue cell-type markers have high utility in predicting primary tissue cell-type annotations. However authors must discuss whether top-ranked MetaMarker gene are consistent with our previous knowledge. For example, it is expected that GABEergic neurons display SLC32A1, GAD1, and GAD2 as high-ranked marker, MetaMakrers in dividing progenitor possibly contain TOP2A and MKI67. If these well-defined markers are not listed as high rank, authors should discuss why these genes are not defined as top-ranked meta markers.

* Overall, the research strategy is interesting, but each dataset seems not to be fairly preprocessed (Some dataset combined multiple protocols, whereas others separated). The current manuscript remains concern that this unfair preprocessing leads to artifacts of gene co-expression patterns. 

Minor concerns:

* Provide top-ranked MetaMarkers as supplemental table. This is very important information in this study.

Reviewer #3 (Alessandro Fiorenzano, signs review): In the manuscript titled "Preservation of Co-expression Defines the Primary Tissue Fidelity of Human Neural Organoids," the authors present a meta-analytic pipeline. This approach quantifies gene-level preservation of primary tissue co-expression, enabling cell-type specific quantifications within brain organoids. The analysis encompasses 2.95 million cells from primary brain tissue across 50 datasets, and 1.63 million cells from neural organoids across 130 datasets, each differentiated using distinct patterning protocols. This extensive examination aims to establish robust cell-type specific markers and co-expression patterns, facilitating the analysis and validation of whether in vitro brain organoid models faithfully replicate authentic molecular identities in culture. Additionally, the authors offer an R package for the convenient download of the primary tissue reference co-expression network. This tool facilitates the analysis of new neural organoid data using straightforward, meaningful, and efficient statistical methods.

The manuscript introduces novel elements that address a critical concern in the field of "human biology," particularly within the field of human brain organoid generation. This issue revolves around the absence of well-established human standards, presenting a substantial challenge in the context of advancing 3D modeling (KB Jensen et al., Stem Cell Reports 2023). The inherent inaccessibility of the human brain hampers direct comparisons and validation of results, potentially leading to the production of unreliable data lacking essential controls. The authors' efforts to develop a pipeline that can be utilized across different laboratories may represent an important tool for validating molecular identity and conducting quality checks for ongoing in vitro human stem cell brain models. This commendable endeavour has the potential to significantly contribute to the standardization and reliability of research in the brain organoid field.

Only minor points need to be addressed to make the manuscript suitable for publication. 

Please see a list of minor comments below:

1) The manuscript would benefit and provide stronger conclusions by integrating two relevant studies currently missing in the manuscript: La Manno et al., Cell, 2016, and Birtele et al., Development 2021. The authors should include these two studies as they are relevant, using human fetal mesencephalic tissue to compare hPSC differentiation into dopamine neurons and midbrain organoids.

2) To facilitate the use of the proposed meta-analytic pipeline, the authors should provide a detailed list of primary tissue single-cell datasets in supplementary, including tissue source, the number of cells analyzed, and quality control. Similarly, the authors should provide a list of brain organoid datasets, specifying their regional brain identity and quality control for each dataset analyzed.

3) While efforts are made to model the human brain, the main datasets present are derived from the mouse or, at least, from the differentiation of mouse stem cells, representing a valuable source of information. It would be helpful for the manuscript if the authors could discuss the possibility of meta-analytic approaches to study evolutionary differences and similarities between brain regions of different species. Additionally, shedding light on how animal models can be exploited as an information source to decipher aspects of the human brain would underscore the broad applicability of meta-analytic approaches.

Reviewer #4: In this paper, Werner and Gillis described their analytical framework based on marker identification and co-expression analysis, to estimate fidelity of the molecular programs driving cell type heterogeneity of primary developing human brains in neural organoids. The authors demonstrated their approach with unannotated primary data sets. When applying the analysis to neural organoid scRNA-seq data, the authors reported generally substantially weaker preservation, as well as high variability, of cell type marker co-expression patterns in organoids, suggesting potential limitations of the current neural organoid protocols.

In general, I think the approach described in the manuscript is interesting, scalable, and can be indeed applicable to different systems to derive a general view of system consistency. Meanwhile, I also have several critical concerns, not necessarily on the approach itself, but also how much the results obtained with the approach in comparing neural organoids with primary developing human brains are informative and useful to the neural organoid communities.

1. My first technical concern is the reliability of the methods against undesired technical/biological confounding factors. While gene co-expression is truly a great metric of how close two genes interplay in biological processes, it can also be affected by many different technical factors (especially when looking at co-expression on single-cell level), as well as biological factors which could confound the interpretation. Technical factors include cell number, data coverage, etc., and the potentially confounding biological factors include different cell type composition and proportions in different data sets, expression levels of different genes and so on. The authors indeed spent effort to demonstrate that those factors would not affect the results, which I very much appreciate. Still, they are not yet convincing enough to me.

 1) For example, the authors show that in primary MetaMarkers per cell type show the highest average expressions in the expected cell type generally regardless of brain regions (Suppl. Fig. 2). This, however, doesn't mean that the co-expression of those genes are of consistent level in different brain regions. The authors should directly clarify it, for example, by subseting into different brain regions and checking the co-expression patterns in different brain regions are consistent and of high level. In fact, even the showed results are not perfect (e.g. in many 2nd trimester primary data the GABA- and glutamatergic neuron markers don't show consistent pattern in all regions), and that is even just the average of markers instead of individual markers.

 2) The authors also provided the correlation between zero-padding of genes (which should be somehow negatively correlated with overall gene expression and sequencing depth) and preserved co-expression, which I also appreciate. This analysis, however, focused on how those factors could affect the gene-set level comparison, and skipped their potential influence on the overall comparison between data sets. This should not be very difficult to address, by correlating the overall co-expression preservation with the technical factors such as cell number per data set and average detected transcript numbers and so on.

2. Another technical concern I have is the interpretability of the method. While the proposed analysis can indeed evaluate the overall fidelity of a system compared to the reference, it seems to lack of the capacity to identify potential biological processes which may contribute to the deficit of the system, which can potentially enlighten researcher in the field to improve the system. The authors did explore the possibility by introducing the gene-set level analysis. I appreciate such effort. Some of the reported findings also make sense and are interesting, for example, the lack of preservation on angiogenesis and immune functions. However, this analysis is also largely limited. As the authors mentioned, the preservation of co-expression is negatively linked with whether a cell type is generated or not in the organoid. As non-neural cells including endothelial cells and immune cells are not expected to be there, both the lack of co-expression preservation of the non-neural cell type markers and the reported enriched GO terms don't actually provide any more insight. Indeed, most researchers in the field care more about how to make the expected cell types (e.g. neural progenitor cells, neurons and glial cells) close enough to their primary counterpart, instead of to make the other non-neuroectoderm-derived cells altogether. On top of that, as the co-expressions were estimated with all cells, it is also difficult to tell which cell type are affected the most.

3. This study largely focuses on high-level cell type heterogeneity, which is of course very important. However, the equally, if not more, important aspect is whether the generated cell types are of the expected regional and subtype identities. The proposed method and the reported results don't seem to have the capacity to address this, which largely limit the value of this study to the neural organoid community. The authors should explain and clarity.

4. So far, there are already quite some studies looking into fidelity of neural organoids recapitulating developing human brains. Some (e.g. Camp et al. 2015, Velasco et al. 2019, Uzquiano et al. 2022) focus on the recapitulation and consistency of neural organoids to primary development, and others focus on differences between the two systems (e.g. Bhaduri et al. 2020, Vértesy et al. 2022). There is also a more recent study in biorxiv (He et al. 2023) which also integrated many scRNA-seq data of neural organoids and compared with primary developing human brain atlases similar to this study, but with more complicated analysis on cell type matching, differential expression analysis, etc. All those studies somehow highlighted some important biological processes happened to certain cell types (mostly neural progenitors and neurons). I don't think this study should provide as detailed information as the above mentioned studies, given that the proposed method in this study should in principle be more ease-to-use and scalable as its pros. Still, I would expect the authors to refer to and acknowledge those previous studies, and try to check whether the previous finding can be confirmed with their method.

5. The primary reference atlas data sets used in the study are largely biased to the neocortex and telencephalon, especially the second trimester data. Many "multi-region" data (e.g. Bhaduri et al.), while indeed include multiple brain regions, have huge difference on measured cell numbers in different brain regions. Together with my first concern, it makes me wonder how representative and robust the reference co-expression pattern is to cell types in different brain regions. And regarding the organoid data sets, I also notice that some high-quality data from well-known labs are not included (e.g. Pellegrini et al. 2020 from Lancaster lab, Eichmüller et al. 2022 and Vértesy et al. 2022 from Knoblich lab, Kanton et al. 2019 and Fleck et al. 2022 from Treutlein lab). I'm therefore curious about the standard of data selection and whether they are comprehensive and representative enough.

6. A minor concern: I notice that the authors used 'cerebral' in supplementary table 1 to represent organoids derived using unguided protocols. This doesn't seem to be the ideal term, as the unguided protocols are able to generate many non-cerebral, e.g. non-telencephalic neurons. Using "brain" or "unguided" will be much more proper, in my opinion.

---

## [Decision Letter · Decision Letter 2]

25 Jun 2024

Dear Dr Gillis,

Thank you for your patience while we considered your revised manuscript "Preservation of co-expression defines the primary tissue fidelity of human neural organoids" for publication as a Research Article at PLOS Biology. Your revised study has been evaluated by the PLOS Biology editors, the Academic Editor and the original reviewers.

In light of the reviews, which you will find at the end of this email, we would like to invite you to revise the work to thoroughly address the reviewers' reports.

As you will see, while the reviewers think this revised version of the manuscript is improved, and Reviewers 3 and 4 suggest we accept the study, Reviewer 2 still has a number of important concerns which we think should be carefully addressed before publication. After discussing reviewer 2's comments with the Academic Editor and with the other reviewers, we think that addressing Reviewer 2's comments will require additional bioinformatic analyses as well as textual changes. We would like to emphasize that Reviewer 2's comments about protocol classification is important as it will be essential for the field that the methods being compared are properly described and that like-for-like comparisons are done. We would suggest that you revisit how the datasets were assigned to particular groups, and clearly explain the classification procedure in the manuscript.

Given the extent of revision needed, we cannot make a decision about publication until we have seen the revised manuscript and your response to the reviewers' comments. Your revised manuscript is likely to be sent for further evaluation by all or a subset of the reviewers.

**IMPORTANT - SUBMITTING YOUR REVISION**

*Re-submission Checklist*

*Published Peer Review*

*PLOS Data Policy*

*Blot and Gel Data Policy*

Sincerely,

Suzanne

Suzanne De Bruijn, PhD, 

Associate Editor

PLOS Biology

sbruijn@plos.org

REVIEWS:

Reviewer #2: In the revised manuscript, authors re-organize the collected meta data, provide additional key information, and re-perform the co-expression analyses. The manuscript is improved, but still contains several critical concerns. Therefore, before the publication, these concerns should be revised or addressed by additional data.

* Supplementary Table 1 is improved, but it is unclear why Shi et al. datasets were separated into four groups. This GEO repository contains 3 biological replicates per group, and these replicates should be processed separately, since each replicate could not generate the organoid with the same cell composition and may affect the co-expression. Is it technically difficult to separate this data by biological replicate?

* I still have major concerns about their protocol classification. For example, authors classified Kadoshima et al. protocol as directed dorsal forebrain, and Eiraku et al. protocol as directed cortex. Kadoshima's protocol is an improved version of Eiraku's, and the most of medium compositions are same (See Kadoshima et al. PNAS, 2013). Esk et al. used Lancaster et al. 2017 protocol. However, authors classified Esk et al. and Lancaster et al 2017 as different groups. In addition, Pasca et al. 2015 and Sloan et al. 2018 used the same protocol, but were classified into different groups. Authors may classify protocols based on the description of the publications, but the most of publications might not clearly distinguish "dorsal forebrain" and "cortical". Although I appreciate authors' efforts to re-organize the collected datasets, authors should pay more attention to their protocol classification scheme, since authors discussed the variability of the "directed cortical" protocols in the main text (Fig. 3, Supp Fig 4). The interest of the most of biologists is how the differences of the protocols affect their fidelity. Since authors submitted the manuscript to a biology-oriented journal rather than computational biology-related journals (e.g. PLoS computational biology), the protocol must be categorized by reasonable and acceptable way for biologists. 

* After identification of MetaMarkers, authors addressed "how well these markers capture organoid temporal and regional (protocol) variation". Then, authors performed AUROC quantification using top 100 markers from each primary dataset, and obtained consistent performance regardless of primary dataset or organoid timepoint (except for neuronal progenitor). This is reasonable and understandable. In contrast, to address regional (protocol) variation, authors performed PCA, and used PC1 weights to assess the variation. This PCA-based analysis is complicated, and may lead misinterpretation. I think that more explanation or additional analyses are needed.

- For example, PC1 weights are highly variable among organoid protocols (Supp. Fig. 3C). Even within directed dorsal forebrain protocols, PC1 weights are quite different. According to their new Supplemental Table 1, the dorsal forebrain protocol is composed of four original protocols, Kadoshima et al, Sloan et al., Qian et al. and Esk et al. Is this due to the difference of original protocols?

- Is PC1 really good assessment measurement for protocol/regional variation? Instead of PCA, how about using top100 markers per primary region as did in Supp. Fig. 3B and Supp. Fig 2? From new Supplemental Table 1, primary tissues data seems to contain the corresponding regions to certain organoid protocols (e.g. ganglionic eminence, thalamus, hypothalamus). Authors identifies top 100 markers for one primary region, and calculate AUROC in the organoid protocols for this region, other regions, and undirected organoid protocols. 

- If the suggested method is technically difficult, please carefully explain the rationale of PCA-based analysis. 

- A word "capture" in the section title is very vague…. Does author claim that broad primary tissue cell-type markers are independent from organoid temporal/regional (protocol) variation?

* MetaMarkers for non-neuronal cell types seem to be biased to endothelial cells and microglia (RAB31, C1QC, C1QB, C1QA, and R2PY12) (Supp Table 3). Therefore, co-expression module AUROC is very high in vascular organoid, neuro-vascular, iPSC-microglia and vascularized undirected organoid. In addition, undirected protocol also can generate mesodermal progenitors (precursor of endothelial cells and microglia) (Ormel et al., Nat Commun 2018), whereas directed protocols cannot. Since many scientists imagine "non-neuronal" cell type as oligodendrocyte or astrocyte, this may lead misinterpretation. Authors should discuss this as limitation of their cell type classification approach in the manuscript, or carefully classify "non-neuronal" cells into each glial cell type. 

* Line 360: "Interestingly, the protocol with the most variable performance for cell-type co-expression across datasets is the directed cortical protocol, which can be classified as an intermediate protocol in terms of directed differentiation between the undirected and region-specific protocols". Is this true? Usually, the direction level of the organoid is controlled by signaling inhibitors or ligands (Bhadiri et al., Nature 2020). Have authors interpreted directed cortical protocols as "intermediate protocols" by the differences of signaling inhibitors/ligands. If so, please more carefully explain why the directed cortical protocols can be examined as "intermediate"? Authors categorized five different protocols (Field et al. 2019, Popova et al., 2021, Revah et al., 2022, Vertesy et al., 2022, Xiang et al., 2017, two authors-developed, Lancaster 2017-based, Eiraku-based, and Pasca-based) into directed cortical. Is it possible that the variability may be due to the variation of protocols rather than the direction level.

* Line 370: "While variability in co-expression performance may reflect compositional differences among neural organoids, the dramatic range in performance across the directed cortical organoid datasets reveals extensive variability in fidelity to primary tissue." & Line 433: "regardless of putative compositional variation across organoid protocols and/or treatments, we demonstrate the vast majority of organoids from all sampled protocols consistently fail to preserve glutamatergic or GABAergic primary tissue co-expression at a level comparable to primary tissue". For these conclusions, their results are insufficient. As also discussed in the manuscript, the AUROC may be affected by the compositional variation of cell types. Since the dorsal cortical organoid group is a mixture of multiple protocols, it is possible that the cell composition is highly different among protocols compared to other groups. To support their conclusion, authors should provide data showing that the AUROC is not or weakly dependent on the cell composition (e.g. correlation between cell type ratio and AUROC, as did in Supp Fig. 9) or tone down this conclusion. 

* Line 438: "Organoids across protocols additionally exhibit near-zero preservation of non-neuronal primary tissue co-expression, suggesting neural organoids generally do not produce or produce extremely dysregulated non-neuronal cell-types (Fig. 4B, Supp. Fig. 5)." Due to the strong bias of non-neuronal cell types to endothelial cells and microglia, no preservation of non-neuronal primary tissue co-expression is due to lack of these cell types in the organoid, but not dysregulation of non-neuronal cell types. 

* Section: "Fidelity of finer resolution cell-types through preserved co-expression". This section is very interesting, but hard to read for biologists. Especially, "finer resolution cell type" may be difficult to be understood by biologists. How about changing it to "region-specific cell types"?

* Line 541: "These results highlight preserved primary tissue co-expression of ECM-related genes as a particularly consistent deficit across neural organoids, suggesting that investigations into the signaling between artificial ECM and cells in organoid cultures may be a route forward for general improvements of organoid fidelity." ECM is one of major characteristics of endothelial cells. ECM genes are highly expressed in endothelial cells, but not in other neuronal cell types. Therefore, the poor conservation of ECM genes in organoid is due to the lack of endothelial cells.

* In the temporal variation analysis of the preserved co-expression (Fig. 5), did authors use top 100 or top 10 cell-type markers? Authors described top 100 in main text, whereas Fig 5A described top 10.

* Line 566: "Our approach in predicting organoid annotations in Figure 2 is based on aggregate marker expression and did not produce temporally variable results, whereas our approach here comparing preserved co-expression of the same marker genes does produce temporally variable results.". Authors selected MetaMarkers as markers in broad primary tissue cell types among time points and brain regions (Fig. 2), whereas the analyses in Fig. 5 used markers in each primary dataset. Therefore, there is no surprising about this conclusion. Rather than comparing with Fig. 2, authors should compare the results of Fig. 5 with Supp. Fig. 3. If my understanding is correct, Fig. 5 was assessed by the preservation of co-expression, while Supp. Fig. 3 was evaluated by gene expression levels to define cell type annotation. This is well matched with their final conclusion "This indicates that the co-expression relationships of genes rather than their expression levels better capture temporal variation in developing systems".

* I think that the relationship of temporal variation with co-expression and expression levels is very different among cell types. For example, neuronal progenitors showed the temporal variation in both Fig. 5 and Supp Fig. 3. In contrast, intermediate progenitors, GABAergic neurons, and glutamatergic neurons showed temporal variation in the co-expression preservation, but not in cell type annotation. I think that this is interesting, and authors should discuss these differences among cell types in main text.

* For the comparison with Supp Fig. 3, it is better to add the co-expression analysis for dividing progenitors and non-neuronal cell type in Fig. 5. 

Reviewer #3: identified himself as Alessandro Fiorenzano

"The manuscript is now suitable for publication"

Reviewer #4: 

"I very much appreciate the effort by the authors to address my concern. I hope the authors find my previous comments helpful to improve the manuscript. I think the manuscript is now good enough to be accepted in PLOS Biology."

---

## [Editor Report · Decision Letter 3]

9 Oct 2024

Dear Jesse,

Thank you for your patience while we considered your revised manuscript "Preservation of co-expression defines the primary tissue fidelity of human neural organoids" for publication as a Research Article at PLOS Biology. This revised version of your manuscript has been evaluated by the PLOS Biology editors and the Academic Editor.

Based on our Academic Editor's assessment of your revision, I am pleased to say that we are likely to accept this manuscript for publication, provided you satisfactorily address the following data and other policy-related requests that I have provided below (A-E):

(A) We routinely suggest changes to titles to ensure maximum accessibility for a broad, non-specialist readership. In this case, we would suggest the following title suggestions as follows:

“Meta-analysis of single-cell RNA sequencing data from human neural organoids identifies a preserved co-expression framework as a quality control metric for their ability to recapitulate primary tissue"

OR 

“Meta-analysis of single-cell RNA sequencing data from human neural organoids reveals high variability in the ability of organoid systems to recapitulate primary tissue”

(B) Thank you for already providing the source data for all the relevant figure panels in your Github deposition. Please note that we cannot accept sole deposition of code or the source data in GitHub, as this could be changed after publication. However, you can archive these versions of your publicly available GitHub code or source data to Zenodo. Once you do this, it will generate a DOI number, which you will need to provide in the Data Accessibility Statement (you are welcome to also provide the GitHub access information). See the process for doing this here (https://docs.github.com/en/repositories/archiving-a-github-repository/referencing-and-citing-content). In addition, please also link your code used to generate the figure panels and the preservedCoexp R library in the Zenodo repository.

(C) In addition, I would also be grateful if you could please label the individual files so it is made clear which specific source data files relate to each figure panel, as we require the raw data for the following figures:

Figure 1B-C, 1F, 2A, 2C-E, 3A-B, 3D-E, 4B-G, 5B, 6A-B, S1A-B, S2A-B, S3B-D, S4A-D, S5A-C, S6A-B, S7A-F, S8A-G, S9A-C, S10A-D

(D) Please also ensure that each of the relevant figure legends in your manuscript include information on *WHERE THE UNDERLYING DATA CAN BE FOUND*, and ensure your supplemental data file/s has a legend.

(E) Please ensure that your Data Statement in the submission system accurately describes where your data can be found.

We expect to receive your revised manuscript within two weeks. 

*Published Peer Review History*

*Press*

Kind regards,

Richard

Richard Hodge, PhD

rhodge@plos.org

PLOS

---

## [Editor Report · Decision Letter 4]

24 Oct 2024

Dear Jesse,

On behalf of my colleagues and the Academic Editor, Madeline Lancaster, I am pleased to say that we can accept your manuscript for publication, provided you address any remaining formatting and reporting issues. These will be detailed in an email you should receive within 2-3 business days from our colleagues in the journal operations team; no action is required from you until then. Please note that we will not be able to formally accept your manuscript and schedule it for publication until you have completed any requested changes.

In addition, the editorial team found the new title a bit difficult to parse, so I have taken the liberty of including a minor edit. Please do let me know if you OK with this. 

"Meta-analysis of single-cell RNA sequencing co-expression in human neural organoids reveals their high variability in recapitulating primary tissue”

PRESS

Best wishes, 

Richard

Richard Hodge, PhD

rhodge@plos.org

PLOS
